# Validation of TROPOMI Surface UV Radiation Product

Kaisa Lakkala[1,2], Jukka Kujanpää[1], Colette Brogniez[3], Nicolas Henriot[3], Antti Arola[2], Margit Aun[2,4], Frédérique Auriol[3], Alkiviadis F. Bais[5], Germar Bernhard[6], Veerle De Bock[7], Maxime Catalfamo[3], Christine Deroo[3], Henri Diémoz[8,9], Luca Egli[10], Jean-Baptiste Forestier[11], Ilias Fountoulakis[8], Katerina Garane[5], Rosa Delia Garcia[12,13], Julian Gröbner[10], Seppo Hassinen[1], Anu Heikkilä[2], Stuart Henderson[14], Gregor Hülsen[10], Bjørn Johnsen[15], Niilo Kalakoski[1], Angelos Karanikolas[5], Tomi Karppinen[1], Kevin Lamy[11], Sergio F. León-Luis[13], Anders V. Lindfors[16], Jean-Marc Metzger[17], Fanny Minvielle[3], Harel B. Muskatel[18], Thierry Portafaix[11], Alberto Redondas[13], Ricardo Sanchez[19], Anna Maria Siani[20], Tove Svendby[21], and Johanna Tamminen[1]

[1]Finnish Meteorological Institute, Space and Earth Observation Centre
[2]Finnish Meteorological Institute, Climate Research Programme
[3]Univ. Lille, CNRS,UMR 8518, Laboratoire D'Optique Atmosphérique, Lille, France
[4]University of Tartu, Estonia
[5]Aristotle University of Thessaloniki, Greece
[6]Biospherical Instruments, Inc, San Diego, U.S.
[7]Royal Meteorological Institute of Belgium, Brussels, Belgium
[8]Aosta Valley Regional Environmental Protection Agency (ARPA), Saint-Christophe, Italy
[9]Institute of Atmospheric Science and Climate, ISAC-CNR, Rome, Italy
[10]Physical Meteorological Observatory in Davos - World Radiation Center, Switzerland
[11]LACy, Laboratoire de l'Atmosphère et des Cyclones (UMR 8105 CNRS, Université de La Réunion, Météo-France), Saint-Denis, Reunion Island, France
[12]Atmospheric Optics Group, Valladolid University, Valladolid, Spain
[13]Izaña Atmospheric Research Center (IARC), State Meteorological Agency (AEMET), Spain
[14]Australian Radiation Protection And Nuclear Safety Agency
[15]Norwegian Radiation and Nuclear Safety Authority, Norway
[16]Finnish Meteorological Institute, Meteorological and Marine Research Programme
[17]Observatoire des Sciences de l'Univers de La Réunion, UMS3365 (CNRS, Université de La Réunion, Météo-France), Saint-Denis de la Réunion, France
[18]Israel Meteorological Service
[19]National Meteorological Service, Argentina
[20]Sapienza Università di Roma, Italy
[21]NILU - Norwegian Institute for Air Research, Norway

**Correspondence:** Kaisa Lakkala (kaisa.lakkala@fmi.fi)

**Abstract.** The TROPOspheric Monitoring Instrument (TROPOMI) onboard the Sentinel-5 Precursor (S5P) satellite was launched on 13 October 2017 to provide the atmospheric composition for atmosphere and climate research. The S5P is a sun-synchronous polar-orbiting satellite providing global daily coverage. The TROPOMI swath is 2600 km wide, and the ground resolution for most data products is 7.2x3.5 km$^2$ (5.6x3.5 km$^2$ since 6 August 2019) at nadir. The Finnish Meteorological Institute (FMI) is responsible for the development of the TROPOMI UV algorithm and the processing of the TROPOMI Surface Ultraviolet (UV) Radiation Product which includes 36 UV parameters in total. Ground-based data from 25 sites located in arctic, subarctic, temperate, equatorial and antarctic areas were used for validation of TROPOMI overpass irradiance at 305, 310, 324 and 380

nm, overpass erythemally weighted dose rate / UV index and erythemally weighted daily dose for the period from 1 January 2018 to 31 August 2019. The validation results showed that for most sites 60–80% of TROPOMI data was within ±20% from ground-based data for snow free surface conditions. The median relative differences to ground-based measurements of TROPOMI snow free surface daily doses were within ±10% and ±5% at two thirds and at half of the sites, respectively. At several sites more than 90% of cloudfree TROPOMI data were within ±20% from ground-based measurements. Generally median relative differences between TROPOMI data and ground-based measurements were a little biased towards negative values (i.e. satellite data < ground-based measurement), but at high latitudes where non-homogeneous topography and albedo/snow conditions occurred, the negative bias was exceptionally high, from -30% to -65%. Positive biases of 10–15% were also found for mountainous sites due to challenging topography. The TROPOMI Surface UV Radiation Product includes quality flags to detect increased uncertainties in the data due to heterogeneous surface albedo and rough terrain which can be used to filter the data retrieved under challenging conditions.

## 1 Introduction

The Tropospheric Monitoring Instrument (TROPOMI) is a nadir-viewing imaging spectrometer measuring in the ultraviolet, visible, near-infrared, and the shortwave infrared wavelengths onboard the Sentinel-5 Precursor (S5P) polar-orbiting satellite. The S5P was launched on 13 October 2017 as part of the EU Copernicus programme to monitor atmospheric composition with nominal life time of seven years. The mission is a cooperative undertaking between the European Space Agency (ESA) and the Netherlands. The S5P satellite is on a sun-synchronized afternoon orbit with an ascending node equatorial crossing at 13:30, which provides global daily observations of the sunlit part of the Earth for air quality and climate applications. The S5P is the first Copernicus mission dedicated to atmospheric observations and it will be complemented by Sentinel 4 with geostationary orbit and Sentinel 5 on sun-synchronous morning orbit with planned launches in the coming years. The TROPOMI Level 2 data products include information of aerosols, carbon monoxide, clouds, formaldehyde, methane, nitrogen oxide, sulphur dioxide, ozone and surface ultraviolet (UV) radiation. Other products are generated within the Copernicus ground system, while the surface UV radiation is generated through the Finnish Sentinel collaborative ground segment.

Solar UV radiation at short wavelengths (280-400 nm) is harmful for the whole ecosystem including humans, animals, plants, aquatic environments and materials (e.g., EEAP, 2019, and references therein). For humans the well known harmful effects of UV radiation are sunburns and other skin problems, increased risk of skin cancer and cataract, premature aging of the skin and weakening of the immune system (EEAP, 2019). On the other hand UV radiation initiates vitamin D production in the skin (Webb, 2006) and has many more positive effects (Juzeniene and Moan, 2012). The ozone layer in the stratosphere protects the Earth from the most dangerous UV wavelengths by absorbing the shortest part of the spectrum. In the late 1970s the ozone layer was found to decrease at an alarming speed above Antarctica (Farman et al., 1985; WMO, 1990). Later, also

in the Arctic the total ozone was found to decrease in the spring and ozone trends turned to negative at mid-latitudes (WMO, 1999). The international Montreal Protocol was signed in 1987 to protect the ozone layer by phasing out the production of ozone-depleting substances (ODS). As a result, the ozone layer is now starting to recover (WMO, 2018). However, the removal process of ODS will take several decades and UV levels at the ground will therefore remain elevated for the foreseeable future (Petkov et al., 2014; Fountoulakis et al., 2020b).

Ground-based UV monitoring started to increase in the late 1980s to respond to the concerns about increased surface UV levels due to the depleting ozone layer (Solomon et al., 1986). However, the ground-based UV monitoring network is sparse from a global point of view and many places are not covered. The advantage of retrievals from space is that satellites provide global coverage of biologically effective UV parameters. The disadvantage is that for polar-orbiting satellites there is only one overpass per day for most sites. However, daily doses can be estimated using combination of radiative transfer calculations and measurements from satellite instrument during the overpass (e.g., Kalliskota et al., 2000; Tanskanen et al., 2007).

The Finnish Meteorological Institute is responsible for the development, processing and archiving of the TROPOMI Surface UV Radiation Product, which continues UV records started by NASA Total Ozone Mapping Spectrometer (TOMS) instrument in 1978 (Eck et al., 1995; Krotkov et al., 2001) and followed by the Dutch-Finnish Ozone Monitoring Instrument (OMI) onboard NASA's Aura satellite launched in 2004 (Levelt et al., 2006; Tanskanen et al., 2006). Compared to the preceding instruments, TROPOMI has an increased spatial resolution with a swath of 2600 km including 450 across-track pixels. The ground resolution of the UV product was 7.2x3.5 km$^2$ at nadir until 6 August 2019, and is now 5.6x3.5 km$^2$, while the OMI pixel size was 13x24 km$^2$ at nadir. The TROPOMI Surface UV Radiation Product responds to the increasing need of information regarding the troposheric chemistry and biologically active wavelengths of the solar spectrum reaching the surface. In this paper, overpass irradiances at 305, 310, 324 and 380 nm, overpass erythemally weighted dose rates / UV index and daily doses are validated against well maintained and calibrated ground-based spectroradiometer, broadband and multichannel radiometer measurements from 25 sites.

## 2  Data

### 2.1  TROPOMI surface UV radiation product

The TROPOMI surface UV algorithm is explained in detail in Lindfors et al. (2018) and Kujanpää et al. (2020). It is based on the heritage of the surface UV algorithms for the TOMS (Eck et al., 1995; Krotkov et al., 2001; Herman et al., 2009), the OMI (Levelt et al., 2006; Tanskanen et al., 2006; Arola et al., 2009) and the offline UV product (OUV) of the EUMETSAT Satellite Application Facility on Atmospheric Composition Monitoring (AC SAF) (Kujanpää and Kalakoski, 2015). Satellite surface UV products are based on radiative transfer modelling using as main inputs: solar zenith angle (SZA), total ozone column, cloud optical depth, aerosol optical properties, surface pressure and surface albedo. For the TROPOMI product, the VLIDORT radiative transfer model (Spurr, 2006) is used for the radiative transfer calculations.

The TROPOMI UV algorithm is based on two pre-computed lookup tables (LUT) in order to save computing time compared to explicit radiative transfer calculations. The first LUT is used to retrieve the cloud optical depth from the measured 354

nm reflectance using SZA, viewing zenith angle, relative azimuth angle, surface pressure and surface albedo as other inputs. Details on the cloud optical depth retrieval can be found in Sect. 3.3 of Lindfors et al. (2018). The measured 354 nm reflectance together with the angles and surface pressure are obtained from the TROPOMI L2 aerosol index (AI) product (Stein Zweers,

2018) as they are used for the calculation of the AI product. The LUT was pre-generated by radiative transfer calculations. The reflectance at 354 nm was calculated using different combinations of cloud optical depth, SZA, viewing zenith angle, relative azimuth angle, surface pressure and surface albedo. The outcome is a LUT from which the cloud optical depth can be retrieved when all other input parameters are known. For radiative transfer calculations, a homogeneous water cloud is considered at 1-2 km height in the atmosphere. Thus, the retrieved cloud optical depth can be considered to be an effective optical depth for

the whole satellite pixel which best corresponds to the measured 354 nm reflectance. 3D effects due to partial cloudiness are ignored.

The surface albedo is obtained from the surface albedo climatology generated for the AC SAF OUV product (Kujanpää and Kalakoski, 2015) which is provided on a 0.5deg x0.5deg latitude-longitude grid. It uses the monthly minimum Lambert equivalent reflectivity (MLER) climatology (Herman and Celarier, 1997) for regions and time periods with permanent or negligible

snow/ice cover, while elsewhere a climatology from Tanskanen (2004) is used, which better captures the seasonal changes in the surface albedo during the snow/ice melting and formation periods. The following data sets were used to determine the regions and time period with permanent or negligible snow/ice cover: Northern hemispheric monthly snow cover extent data (Armstrong and Brodzik, 2010) from the International Satellite Land-Surface Climatology Project, Initiative II (ISLSCP II) (Hall et al., 2006) together with the monthly masks of maximum sea ice extent derived by the National Snow and Ice Data Cen-

ter (NSIDC) from the sea ice concentrations obtained from passive microwave data (Cavalieri et al., 1996). The climatology of Tanskanen (2004) is calculated from TOMS 360 nm Lambertian Equivalent Reflectivity (LER) time-series 1979-1992 using the moving time-window method presented in Tanskanen et al. (2003). The data is available in a 1deg x 1deg latitude-longitude grid from http://promote.fmi.fi/MTW_www/MTW.html.

The second LUT stores the irradiances and dose rates as a function of total ozone column, surface pressure, surface albedo,

cloud optical depth and SZA. The irradiances and dose rates are obtained by Lagrange polynomial interpolation using the total ozone column from the offline version of the TROPOMI L2 total ozone column product (Garane et al., 2019). Surface albedo and pressure are the same as in the first step. The SZA is either the overpass time value or calculated for the solar noon time. A post-correction for the effect of absorbing aerosols is applied. The correction follows the approach developed earlier for the OMI algorithm (Arola et al., 2009). It is based on aerosol absorption optical depth (AAOD), which is taken from the monthly

aerosol climatology by Kinne et al. (2013). The correction factor and its dependence on AAOD was first suggested by Krotkov et al. (2005) and applied in Arola et al. (2009). The correction for erythemal and vitamin D synthesis weightings is the same as for the 310 nm irradiance. A correction for the variation in Sun-Earth distance is also applied in the post-processing step.

The TROPOMI L2 UV product (Kujanpää, 2020) contains 36 UV parameters in total (Table 1), including irradiances at four different wavelengths and dose rates for erythemal (Commission Internationale de l'Eclairage, 1998) and vitamin D synthesis

(Bouillon et al., 2006) action spectra. All parameters are calculated for overpass time, solar noon time, and for theoretical clear sky conditions with no clouds or aerosols. Daily doses and accumulated irradiances are also calculated by integrating over

the sunlit part of the day. As the cloud optical depth is retrieved at the overpass time, the uncertainties in the final cumulative product (daily dose and accumulated irradiances) increases especially for changing cloudiness. For rapidly changing cloudiness the effect is seen also in noon parameters. In addition to UV parameters, quality flags related to the UV product and processing are generated (Kujanpää, 2020). The processing quality flags are a standard set included in all TROPOMI L2 products while the product quality flags are specific to the surface UV product. A continuous overall quality value number (UVQAV $\in$ [0,1], over 0.5 representing the most reliable data) computed from the product quality flags indicates increasing product quality and can be used together with the quality flags to filter out problematic data.

The level 2 data are stored in netCDF-4/HDF5 format. One file is *ca.* 250 MB (190 MB before 6 Aug 2019) in size. UV product version 1.02.02 was used for the current study. The input total ozone and aerosol index files were collected from the reprocessed and offline data in order to construct as homogeneous a time-series as possible. However, the total ozone product version varies from 1.01.02 (starting from 7 Nov 2017) to 1.01.05 (15 Apr 2018) to 1.07.07 (30 Apr 2018) while the aerosol index product version goes from 1.00.01 (7 Nov 2017) to 1.02.02 (15 Apr 2018) to 1.03.01 (30 Apr 2018) to 1.03.02 (27 Jun 2019). Changes in version numbers do not significantly impact the surface UV product. However, there are signs of degradation in the UV solar irradiance measurement of TROPOMI (Rozemeijer and Kleipool, 2019). We do not see any trend in our cloud optical depth retrievals using the 354 nm reflectance, but further analysis is needed in any UV trend study.

To facilitate the validation work, ground station overpass text files containing the UV parameters and supporting input and quality flag data were extracted from the large L2 files. The extractor (version 1.02.00) computes the great-circle distance between the ground station and TROPOMI pixel centre coordinates using the haversine formula and the Earth radius at the ground station coordinates. When the great-circle distance is smaller than a pre-defined limit, here set to 10 km, the data for the TROPOMI ground pixel are stored. No interpolation between the ground pixels is performed.

**Table 1.** TROPOMI surface UV parameters

| |
| --- |
| Overpass and solar noon time irradiance at 305, 310, 324 and 380 nm [W/m$^2$/nm] |
| Overpass and solar noon time dose rate for erythemal and vitamin D synthesis action spectra [W/m$^2$] |
| Daily accumulated irradiances at 305, 310, 324 and 380 nm [J/m$^2$] |
| Daily dose for erythemal and Vitamin D synthesis action spectra [J/m$^2$] |
| Overpass and solar noon time UV index (dimensionless) |
| + all parameters for clear sky conditions (no clouds nor aerosols) |
| + quality flags (UV product and processing flags, and UV quality value (UVQAV)) |

## 2.2 Ground-based reference data

The TROPOMI surface UV radiation product is compared with ground-based UV measurements. The location and characteristics of the sites are shown in Fig. 1 and Table 2 in which they are listed from North to South. The sites were chosen to represent different latitudes, longitudes and topography. The sites are located in arctic, subarctic, temperate, equatorial and antarctic areas including inland, coastal and mountainous stations. At few stations, snow occurs during some period of the year. Ground-based UV measurements are performed using spectroradiometers, broadband and multiband radiometers. The instrumentation and its characteristics are shown in Tables 3 and 4. Many of the spectroradiometers have participated in on-site quality assurance of spectral solar UV measurements performed by the traveling reference spectroradiometer QASUME (Quality Assurance of Spectral UV Measurements in Europe) since 2002 (Gröbner et al., 2005). The average offset of all instruments is within ±5% from the reference instrument with a diurnal variability typically less than 5%. The reports of the site visits can be found at https://www.pmodwrc.ch/en/world-radiation-center-2/wcc-uv/qasume-site-audits/ and the comparison results of the latest QA-SUME comparisons are shown in Table 3. In addition, estimates of expanded uncertainties of ground-based measurements are shown, when available, in Tables 3 and 4. The expanded uncertainties of spectroradiometers and broadband / multiband radiometers are less than 6% and less than or equal to 9%, respectively.

The Norwegian UV Monitoring Program includes UV measurements at 9 sites throughout Norway. It is a cooperation between the Norwegian Radiation and Nuclear Safety Authority (DSA), Norwegian Institute for Air Research (NILU) and the University of Oslo. Four sites were chosen for this study based on their latitude and topography. Ny-Ålesund is the northernmost site and located in Svalbard. Measurements from the GUV-instrument reveals snow cover typically from mid of September to early July (albedo >0.2). The seasonal maximum albedo is 0.8, but during the later years the albedo is now 0.5-0.6. Andøya is located at the tip of a long island, locally influenced by snow in winter and spring. The sea around the site is usually open. Finse is a mountain village at an altitude of 1200 m, close to the Hardanger-Jøkulen glacier. Measurements from the GUV instrument reveal snow cover typically lasting from 20 September to mid of July (albedo >0.2), but the timing of the melting season may be shifted by ±1 month (2015 and 2018), interspersed with periods with wet snow (end of April 2019). The maximum albedo exceeds 0.90. Blindern is located at the suburban area of the city of Oslo. At all sites, the cloudiness is characterized by rapidly moving clouds. The network is equipped with GUV multifilter radiometers which measure UV irradiance at five channels as one minute averages. The data is used to retrieve the UV index and UV dose rates using several action spectra (Bernhard et al., 2005a; Johnsen et al., 2002, 2008) and is freely available at https://github.com/uvnrpa/. The quality assurance of the network includes transfer of the absolute calibration using a regularly calibrated traveling reference. The data is corrected for drift and for angular dependency. Intercomparisons of UVI against the QASUME reference (2003, 2005, 2009, 2010, 2014, 2019) show an interquartile range within ±5% for all GUV instruments and campaigns performed within the period 2003-2019.

The FMI performs spectral UV measurements with Brewer spectroradiometers in the South of Finland in Helsinki and in the North in Sodankylä. The spectral time series of Sodankylä is one of the longest in the Arctic (Lakkala et al., 2003). The site in Helsinki is located in the vicinity of the city centre, but characterized by urban green area. The measurements are performed at the roof of the FMI main building and the horizon is free except in the North side. The weather is characterized by convective

cloudiness in summer afternoons and humid winters. UV measurements in Sodankylä are part of the research infrastructure of the Arctic Space Centre. The research centre is located 5 km from the village by the river Kitinen and surrounded by swamps and boreal forest. Snow occurs from October to April/May. Temperatures can reach -40°C and +30°C in winter and summer, respectively. The Sun is below the horizon for a couple of weeks during winter, and stays above the horizon during a couple of weeks around mid-summer. The FMI Brewer spectroradiometers are calibrated every second or third month using 1 kW lamps in the laboratory (Lakkala et al., 2016). The primary calibration lamps are calibrated yearly at the National Standard Laboratory MIKES-Aalto (Heikkilä et al., 2016; Kübarsepp et al., 2000). The quality assurance of the measurements includes corrections for temperature dependence and cosine error (Lakkala et al., 2008; Mäkelä et al., 2016; Lakkala et al., 2018) and data are submitted to the European UV data base (Heikkilä et al., 2016). Data is regularly compared to the QASUME reference and differences of less than 6% have been found for wavelengths > 305 nm (Lakkala et al., 2008).

The Royal Meteorological Institute of Belgium operates two Brewer spectrophotometers on the roof of its building at Uccle, a residential suburb of Brussels about 100 km from the shore of the North Sea. The climate is influenced by the Gulf stream with mild winters and warm summers. Cloudiness is most of the time variable. The measurements of the Brewer no. 178 were used in this study. It is a double monochromator Mk III which was installed in September 2001. The raw UV counts are converted to counts per second and corrected for instrument dead time, dark count and temperature. Brewer measurements are calibrated with 50 W tungsten halogen lamps on a monthly basis and with 1 kW lamps during less frequent but regular intercomparisons. The instruments were also compared with the traveling QASUME unit in 2004 (Gröbner et al., 2006a).

The Laboratoire d'Optique Atmosphérique (LOA) performs spectral UV measurements with Bentham spectroradiometers at three French sites, in metropolitan and overseas regions (Brogniez et al., 2016). The first site, Villeneuve d'Ascq (VDA), is a semi-urban site located in a flat region of the North of France close to Lille. It is characterized by an oceanic midlatitude climate (warm summers, mild humid winters). The second site, Observatory of Haute-Provence (OHP), is a rural mountainous site located in the French Southern Alps. It is characterized by a mountainous Mediterranean climate (warm summers, harsh winters). The third site, Saint-Denis (OPA) is a coastal urban site located on the Moufia campus in the small mountainous island of La Réunion in the Indian Ocean. This environment leads to frequent occurrence of orographic clouds forming in early afternoon especially in summer. OPA is characterized by a tropical climate (hot-humid summers, mild-warm winters). At the tropical site UV radiation level in summer is much higher around noon than at the two metropolitan sites due to a higher sun elevation and a lower total ozone column. Note that, at VDA and OHP sites, absorbing aerosols are present, and need to be accounted for in satellite UV algorithms (Arola et al., 2009). Due to its oceanic and mountainous surroundings, OPA is a challenging site for satellite validation, since there might be a large spatial variability of cloud cover and surface type in a satellite pixel. The three instruments are affiliated with NDACC (Network for the Detection of Atmospheric Composition Change), thus to meet the requirements of this network they are calibrated every 2-4 months with 1kW lamps traceable to National Institute of Standards and Technology (NIST) and the measurements are corrected from wavelength misalignment and cosine response. Following Bernhard and Seckmeyer (1999), the expanded uncertainties (k=2) are 5.3% at VDA and OHP and 5% at OPA. At OHP and OPA global irradiance measurements are available every 15 min. At VDA, scans are performed every 30 min. Spectroradiometer's data have been already used for OMI validation (Buchard et al., 2008; Brogniez et al., 2016).

Central European mountainous sites are Davos in Switzerland and Aosta in Italy. Both sites are located in the Alps: Aosta (570 m a.s.l.) being located in a large valley floor with a wide field of view, surrounded by mountains (as high as 3500 m a.s.l.), and Davos a mountainous site stretching from around 1500 m a.s.l. to just above 3000 m. a. s.l. in altitude. UV measurements in Aosta are maintained by ARPA and performed with a Bentham DTMc300 spectroradiometer, which is calibrated on a monthly basis using a set of three 200 W lamps, recently complemented with a setup including two 1 kW lamps. The spectroradiometer is additionally compared to the world calibration reference QASUME every second year. Average differences are generally within $\pm 2\%$, with a diurnal variability below 4%. The whole dataset has been subjected to QA/QC and has been recently re-evaluated and homogenized. The expanded uncertainty for wavelengths above 305 nm and SZAs below 70 degrees is 4%. For larger SZAs and shorter wavelengths the uncertainty is larger, reaching 11% at 300 nm when SZA is 75 degrees (Fountoulakis et al., 2020a). The spectroradiometer is preprogrammed to take measurements every 15 minutes. Weather in Aosta is characterized by warm summer, when convective clouds usually develop along the mountain slopes, and dry winter. Snowfalls occur at the station during some winter days in December-March, while the mountains around the station are covered by snow for most part of the year (October-June).

The measurements of the Swiss site Davos are part of the Physikalisch-Meteorologisches Observatorium Davos, World Radiation Centre (PMOD-WRC). They include spectroradiometer measurements performed with the World reference spectro-radiometer QASUMEII and with the double Brewer #163 using an optimized diffuser (Gröbner, 2003). The spectral solar UV irradiance measurements are traceable to the SI using a set of transfer standards (Gröbner and Sperfeld, 2005). The expanded uncertainty of the spectral solar UV irradiance measurements (k=2) is 1.7% for overcast situations (diffuse sky), and 2.0% for clear sky situations (Hülsen et al., 2016). In addition, 5 broadband UV radiometers (SL1492, SL3860, SL1492, YES010938, KZ 560) measure solar UV irradiance continuously at the site. The average of these measurements is used in this study. The estimated expanded uncertainty (k=2) throughout the year for clear sky measurements of these radiometers is 3.6%, while for all sky conditions the expanded uncertainty is increased to 6.6% due to the increased uncertainty for broken cloud conditions and the corresponding uncertainty of the angular response cosine correction applied to the radiometers. In Davos, mountains limit the field of view so the diffuse radiation is reduced approximately by 5%. There is snow cover from November to March.

Rome and Thessaloniki are both urban sites at the Mediterranean coast. The climate at both sites is characterized by mild humid winters and warm dry summers. Both sites are occasionally under the influence of Saharan dust (Amiridis et al., 2005; Gobbi et al., 2019), which is seen as increased aerosol concentration. The aerosol load can also be increased due to pollution (Fountoulakis et al., 2019). In the summer, most of the days are sunny. In Thessaloniki measurements are performed by the Aristotle University of Thessaloniki with a Brewer MKIII spectroradiometer. The quality assurance of the measurements include: 1 kW lamp calibrations traceable to PTB, temperature and cosine correction (Bais et al., 1998; Garane et al., 2006; Fountoulakis et al., 2017). Detailed information on data quality control and analysis can be found in Fountoulakis et al. (2016a).

The measurements of Rome are maintained by Sapienza Università di Roma and are performed with a Brewer spectrora-diometer. UV irradiance and total ozone content have been measured since 1992 at Rome by the Brewer Mk IV spectropho-tometer No. 067. The overall performance of Brewer 067 has been controlled every 2 years since 1992 by the intercomparison with the traveling standard reference Brewer 017 from International Ozone Services Inc. (IOS), (Siani et al., 2018).The last

calibration was performed in July 2019 and the UV calibration was completed using IOS 1 kW lamp. UV data are processed using cosine and temperature correction. The instrument was also compared with the traveling spectroradiometer QUASUME unit during the UV intercomparison campaign in Arosa (Switzerland) in 2012. UV measurements are taken every 30 minutes.

Measurements at the Israeli sites, Bet-Dagan, Jerusalem and Eilat are maintained by the Israel Meteorological service (IMS). Bet-Dagan station is located in open shrublands near Tel-Aviv metropolis on the coast of the Mediterranean Sea. It is characterized by hot and humid summers and mild winters. The city of Jerusalem is located on the Judean Mountains with hot and dry summers and cold winters. Most of the rain occurs between October and May. Eilat is located on the north coast of the Red Sea surrounded by the mountains of Eilat. The climate there is typical for deserts with hot and arid conditions, the maximum temperature in summer are often over 40°C with constant clear skies conditions between June and September. Winter is also relatively hot with maximum temperatures around 20°C and with an annual average precipitation of 25 mm. UV index is monitored every minute by calibrated Yankee Environmental Systems (YES) UVB-1 radiometers, and the data is saved as 10 minute averages.

The Izaña Atmospheric Observatory is a high-mountain station located on the island of Tenerife (Canary Islands, Spain; 2373 m a.s.l.). The observatory is thus located in the region below the descending branch of the Hadley cell, typically above a stable inversion layer and on an island far away from any significant industrial activities. This ensures clean-air and clear-sky conditions all year. This predominant meteorological conditions of trade wind inversion give rise to the presence of a dense stratocumulus layer of clouds lying below the observatory (García et al., 2016). The surroundings of the observatory is characterized by low bushes and rocks (García et al., 2019).The UV measurements reported are performed with a Brewer no. 183 from the European Brewer Calibration Centre (RBCC-E) maintained by the Spanish State Meteorological Agency (AEMET). The RBCC-E triad is calibrated annually from of 1 KW (NIST traceable) lamps used the observatory facilities (Guirado et al., 2012). The UV response of each instrument is checked regularly used a 200W portable lamp system (Sierra Ramos, 2012). In addition, during the RBCC-E campaign, the travelling reference Brewer no. 185 is compared every year with the QASUME unit from PMOD-WRC (Egli, 2019; Gröbner et al., 2006a). The comparison has been shown to be within 2% with a daily variation of less than 5%. Then, in the Izaña Observatory, the UV measurements of Brewer no. 183 and no. 157 are intercompared with those obtained by the Brewer no. 185 to check its calibration. The difference between Brewer no. 183 used in this comparison and Brewer no. 185 is arround 1%.

The University of La Réunion monitors UV radiation with Kipp&Zonen UVS-E-T radiometers at four sites: Mahé - Seychelles, Antananarivo - Madagascar, Anse Quitor - Rodriguez, and Saint-Denis - Reunion Island. The stations are part of the UV-Indien network. The objective of this network is to monitor and study UV radiation over on the southwestern basin of the Indian Ocean. This region has very few measurements of solar UV irradiance and shows extreme UV Index (UVI) throughout the year. In the context of climate change, this region of the world (southern hemisphere tropics) could be affected by a decrease in ozone and an increase in UVR levels throught the 21st century (Lamy et al., 2019). UV-Indien measurement sites correspond to various environments (seaside, altitude, urban) and are homogeneously distributed throughout the Western Indian Ocean. These radiometers are calibrated every 2 years, either at the WRC Davos Switzerland, or directly from the measurements of the Bentham DM300 spectroradiometer installed on the site of the University of la Réunion Island and managed jointly with the

University of Lille (see the section on the sites of the University of Lille for a description of the Moufia site). The more recent instruments (MAH, ANT and ROD) used the manufacturer's calibration . Raw data are corrected according to the calibration. The calibration coefficient depends on the SZA and the ozone totale column. For the ozone total column, the OMI total ozone column OMTO3 product is used.

The Australian sites, Alice Springs and Melbourne are maintained by the Australian Radiation Protection and Nuclear Safety Agency (ARPANSA). Melbourne is a city of 5 million inhabitants located in the southeastern part of Australia on the shores of Port Phillip Bay. Like all Australian cities, Melbourne is sprawling and has a low population density by world standards. The climate is oceanic with hot summers and mild winters. The weather can change rapidly, especially during summers, due to the location of the city between hot inland and cold southern ocean. Heavy storms and rain associated to cold fronts are

typical during summers, while winters are more stable but cloudy. Measurements in Melbourne are perfomed using a Bentham DTMc300 spectroradiometer. This instrument is calibrated for irradiance twice a year using a 1 kW QTH lamp whose output is traceable to NIST and the wavelength calibration is based on the UV spectral lines of a mercury lamp. Alice Springs was selected to represent inland Australian site. The site is located in the Northern Territory of Australia and it is surrounded by deserts. Summers are extremely hot and dry while winters are short and mild. The average temperatures during summer are

over 30°C, and the minimum temperatures can drop below 10°C during winter. There are typically more than 200 cloud-free sunny days per year in Alice Springs. The UV Index is monitored using a radiometer manufactured by sglux GmbH (Berlin, Germany). The sensor is a hybrid SiC photodiode model UV-Cosine_UVI or ERYCA. A logger records data every minute and the average over ten minutes is calculated during post-processing. The radiometer is exchanged every second year for an equivalent sensor that has been calibrated at the ARPANSA laboratory in Melbourne against the Bentham spectroradiometer.

All data for Alice Springs reported in this paper was collected with a single UV sensor.

    Marambio Base is located on the highest part of the Seymour/Marambio Island, surrounded by the Weddell sea on the north-east side of the Antarctic Peninsula. As a cooperation between the Argentinian National Meteorological Service and the FMI, GUV-radiometer, model GUV-2511, measurements started in 2017. Near real time data is shown in http://fmiarc.fmi. fi/sub_sites/GUVant/ for the last five days. The temperatures at the site are around 10°C in summer and can drop down to

290 -30°C in winter. The soil is frozen and covered with snow most of the year and the Weddell Sea in the East is frozen during the winter, but the coast at Marambio is free from ice the whole year. In the summer heavy cloudiness and fog are common. The station is part of the Global Atmospheric Watch (GAW) program of the World Meteorological Organization (WMO). Two radiometers rotate so that one is measuring at the site while the other is calibrated by Biospherical Instruments, Inc, U.S. and also participates in solar comparisons in Sodankylä, Finland. The expanded (k=2) uncertainty of the GUV measurements in

Marambio is 9% at SZAs smaller than 80° (Lakkala et al., 2020).

    Palmer Station is located on Anvers Island on the west coast of the Antarctic continent. It is a research station of the United States, operated by the U.S. National Science Foundation. UV measurements are performed with a SUV-100 spectroradiometer and are part of the NOAA Antarctic UV monitoring network. The effective albedo at Palmer Station is about 0.8 in winter and 0.4 in summer (Bernhard et al., 2005a). The sea adjacent to the station is frozen during winter and open during summer.

Temperatures can fall below -20°C in winter and can reach up to 10°C in summer. Heavy winds are frequent during winter

time. The quality assurance of the spectroradiometer was described by Bernhard et al. (2005a) and includes comparisons with results of radiative transfer models and measurements of a GUV-511 multi-channel filter radiometer that is deployed next to the SUV-100 instrument. The expanded (k=2) uncertainty for erythemal irradiance measured by the SUV-100 spectroradiometer is 5.8% (Bernhard et al., 2005b).

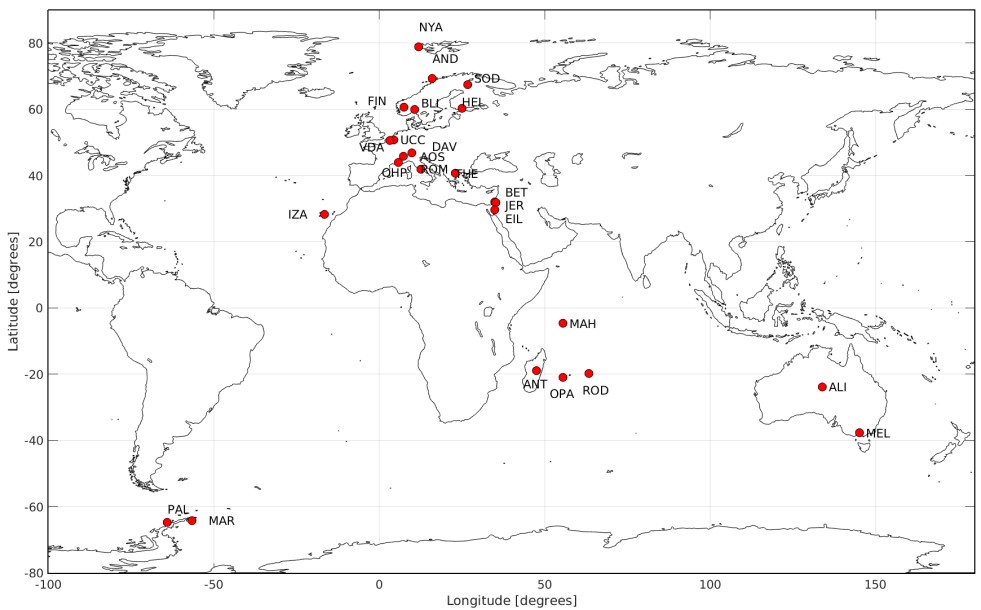

**Figure 1.** Location of ground-based reference sites. See Table 2 for explanation of site acronyms.

**Table 2.** Validation sites ordered according to latitude from North to South.

| Site | site's acronym | Affiliation | Lat., $^{o}$N | Long., $^{o}$E | Elev., m | Characteristics |
|---|---|---|---|---|---|---|
| Ny-Ålesund | NYA | NILU/DSA | 78.924 | 11.930 | 10 | Arctic coast |
| Andøya | AND | NILU/DSA | 69.279 | 16.009 | 380 | Arctic coast |
| Sodankylä | SOD | FMI | 67.367 | 26.630 | 179 | rural, subarctic |
| Finse | FIN | DSA/NILU | 60.593 | 7.524 | 1200 | mountainous |
| Helsinki | HEL | FMI | 60.203 | 24.961 | 43 | urban, subartic coast |
| Blindern | BLI | NILU/DSA | 59.938 | 10.717 | 90 | urban, subarctic coast |
| Uccle | UCC | RMIB | 50.797 | 4.358 | 100 | suburban |
| Villeneuve d'Ascq | VDA | Univ. Lille | 50.611 | 3.140 | 70 | suburban |
| Davos | DAV, DBR, DBB | PMOD-WRC | 46.813 | 9.844 | 1610 | mountainous |
| Aosta | AOS | ARPA | 45.742 | 7.357 | 570 | valley, mountainous |
| Haute-Provence | OHP | Univ. Lille | 43.935 | 5.712 | 688 | rural, mountainous |
| Rome | ROM | Univ. Rome | 41.901 | 12.516 | 70 | urban |
| Thessaloniki | THE | Aristotle Univ. | 40.634 | 22.956 | 60 | urban, Mediterranean coast |
| Bet Dagan | BET | IMS | 32.008 | 34.815 | 31 | rural, shrublands |
| Jerusalem | JER | IMS | 31.770 | 35.197 | 770 | urban |
| Eilat | EIL | IMS | 29.553 | 34.952 | 22 | urban |
| Izana | IZA | AEMET | 28.308 | -16.499 | 2372 | Top of mountain |
| Mahé | MAH | Univ. Réunion | -4.679 | 55.531 | 15 | coast |
| Antananarivo | ANT | Univ. Réunion | -18.916 | 47.565 | 1370 | urban, medium mountain |
| Anse Quitor | ROD | Univ. Réunion | -19.758 | 63.368 | 32 | coast |
| Saint-Denis | OPA, STD | Univ. Lille/Univ. Réunion | -20.902 | 55.485 | 82 | coast, mountainous |
| Alice Springs | ALI | ARPANSA | -23.796 | 133.889 | 550 | desert |
| Melbourne | MEL | ARPANSA | -37.728 | 145.100 | 60 | coast |
| Marambio | MAR | FMI/SMN | -64.241 | -56.627 | 198 | Antarctic coast |
| Palmer | PAL | NSF/NOAA | -64.774 | -64.051 | 21 | Antarctic coast |

**Table 3.** Spectroradiometers used in the study. Data period is from 1 Jan 2018 to 31 Aug 2019, except for the instruments with footnotes. Eryth. act. denotes which erythemal action spectrum is used for retrieving erythemally weighted dose rates and daily doses. 1987 denotes the McKinlay and Diffey (1987) and 1998 the Commission Internationale de l'Eclairage (1998) action spectrum. The average offset from the QASUME reference spectroradiometer for recent comparisons and the expanded uncertainty is given when available. QASUME comparison reports can be found at www.pmodwrc.ch/en/world-radiation-center-2/wcc-uv/qasume-site-audits/. If no publication is linked to the uncertainty, the expanded uncertainty is an estimation calculated by the operator of the instrument. See Table 2 for explanation of site acronyms.

| Site | Instrument | Eryth. act. | Traceability | QASUME average offset; diurnal change | Expanded uncertainty |
|------|-----------|-------------|--------------|----------------------------------------|----------------------|
| SOD | Brewer Mk II #037 | 1987 | MIKES-Aalto | +1% for wl> 310 nm; ±2% (2018) | |
| HEL | Brewer MK III #107 | 1987 | MIKES-Aalto | -1%; ±2–2.5% (2018) | |
| UCC | Brewer MKIII #178 | 1987 | NIST via Kipp&Zonen | -3–(-4)%; 9% [a] (2004) | |
| VDA[1] | Bentham DTMc300 | 1998 | NIST | -3% NDACC comp.(2014) | 5.3%[b] |
| DAV[2] | Bentham QASUMEII | 1998 | PTB | | 1.7 –2% [c] |
| DBR[3] | Brewer MK III #163 | 1998 | PTB | +1%; 4.4% (2019) | 3% |
| AOS | Bentham DTMc300 | 1998 | PTB | -1%; ±3% (2019) | 4% for wl> 310 nm [d] |
| OHP[4] | Bentham DTMc300 | 1998 | NIST | | 5.3%[b] |
| ROM[5] | Brewer #067 | 1998 | NIST via IOS | 5%; ±2% (2012) | |
| THE | Brewer MKIII #086 | 1998 | PTB | | ±5% (1σ uncertainty) [e] |
| IZA | Brewer #183 | 1998 | PTB via PMOD-WRC | ±3% (2019) | |
| OPA | Bentham DTMc300 | 1998 | NIST | -2% (2013) | 5%[b] |
| MEL | Bentham DTMc300 | 1998 | NIST | | 5% |
| PAL | SUV-100 | 1987 | NIST | | 5.8%[f] |

1) 1.1.–15.5.2018, 14.9.2018–31.8.2019, 2) 15.6.-26.10.2018, 21.3-27.3 and 17.5-23.8.2019, 3) 1.1.-26.7.2018, 29.9.2018 - 31.8.2019, 4) 1.1.-31.1.2018, 5) 5.7.2019-26.8.2019
a) Gröbner et al. (2006a), b) Brogniez et al. (2016), c) Hülsen et al. (2016), d) Fountoulakis et al. (2020a), e)Fountoulakis et al. (2016b), f)Bernhard et al. (2005b)

**Table 4.** Broadband and multiband radiometers used in the study and their characteristics. Data period is from 1 Jan 2018 to 31 Aug 2019.The erythemal action spectrum used for retrieving erythemally weighted dose rates and daily doses is Commission Internationale de l'Eclairage (1998) for all other instruments except for BET, JER, EIL and MAR for which it is McKinlay and Diffey (1987). If no publication is linked to the uncertainty, the expanded uncertainty is an estimation calculated by the operator of the instrument. See Table 2 for explanation of site acronyms.

| Site | Instrument | Data frequency | Bandwidth | Traceability | Expanded Uncertainty for UV index |
|------|------------|----------------|-----------|--------------|-----------------------------------|
| NYA | GUV-541 | 1 min ave | 5 channels, FWHM 10nm | PTB via PMOD-WRC | 6.5%[1] |
| AND | GUV-541 | 1 min ave | 5 channels, FWHM 10nm | PTB via PMOD-WRC | 7.2%[1] |
| FIN | GUV-541 | 1 min ave | 5 channels, FWHM 10nm | PTB via PMOD-WRC | 7.1%[1] |
| BLI | GUV-511 | 1 min ave | 5 channels, FWHM 10nm | PTB via PMOD-WRC | 6.6%[1] |
| DBB | average of KZ560, YES010938, SL501A | 10 min ave | broadband | PTB | 3.6–6.6%[a] |
| BET | YES UVB-1 | 10 min ave | broadband 280-320 nm | Kipp&Zonen | |
| JER | YES UVB-1 | 10 min ave | broadband 280-320 nm | Kipp&Zonen | |
| EIL | YES UVB-1 | 10 min ave | broadband 280-320 nm | Kipp&Zonen | |
| MAH | Kipp&Zonen UVS-E-T | 5 min ave | ISO 17166/CIE S007/E-1999 | Kipp&Zonen | 7%[b] |
| ANT | Kipp&Zonen UVS-E-T | 5 min ave | ISO 17166/CIE S007/E-1999 | Kipp&Zonen | 7%[b] |
| ROD | Kipp&Zonen UVS-E-T | 2 min ave | ISO 17166/CIE S007/E-1999 | Kipp&Zonen | 7%[b] |
| STD | Kipp&Zonen UVS-E-T | 5 min ave | ISO 17166/CIE S007/E-1999 | PTB via PMOD-WRC | 7%[c] |
| ALI | sglux ERYCA | 10 min ave | ISO 17166 | NIST via ARPANSA | 8.7% |
| MAR | GUV-2511 | 1 min ave | 5 channels, FWHM 10nm | NIST via BSI | 9%[d] |

1. The latest QASUME comparison in 2019 shows an interquartile range within ±5%.

a) Hülsen et al. (2020), b) Gröbner et al. (2006b), c) provided by PMOD-WRC, d) Lakkala et al. (2020)

## 3 Validation methods

TROPOMI overpass irradiance at 305, 310, 324 and 380 nm, overpass erythemally weighted dose rate, overpass UV index and erythemally weighted daily dose were compared to ground-based measurements. The ground-based data were used as such, as provided by operators, and no conversion between UV index and dose rate was done. The TROPOMI UV parameters are calculated using the erythemal action spectrum from Commission Internationale de l'Eclairage (1998). Most of ground-based measurements used the same action spectrum, while a couple of sites had still in use the action spectrum from McKinlay and Diffey (1987). The effect of using different action spectrum was modeled (results not shown), and they were in agreement with those of Webb et al. (2011). The uncertainty related to the choice of action spectrum was assumed to be less than 1% for low and middle latitudes sites and less than 2% for high latitude sites.

Spectroradiometers measure complete UV spectra and their data were used for the validation of irradiances. Each spectrum was first deconvoluted and then convoluted using a triangular slit of 1 nm at full-width-at-half-maximum using the Shicrivm package (Slaper et al., 1995) freely available at https://www.rivm.nl/en/uv-ozone-layer-and-climate/shicrivm (last visited 19 Mar 2020) as the TROPOMI irradiance is calculated using that standard slit. Data from Palmer were processed with the algorithm described by (Bernhard et al., 2004).

The validation of overpass erythemally weighted dose rate was performed against both spectroradiometer and broadband radiometer measurements. The measurement frequency of these instruments is different: a spectroradiometer may take from 3 to 6 minutes to scan the whole UV range, while a broadband radiometer can measure every second, even if the final product is saved as 1, 2, 5 or 10 minute average. This means that e.g. during changing cloudiness, the cloudiness conditions in the beginning of the spectrum (at short wavelengths) measured by a spectroradiometer may differ from those at the end of the spectrum (at longer wavelengths). The time stamp of spectroradiometer dose rate can differ between sites. For some sites the time stamp is set at the beginning of the spectrum and for some it is set at the most effective wavelength regarding erythemally weighted UV irradiance, at around 308–311 nm. Most of the spectroradiometers measure only 2–4 scans per hour. In order to get enough points between satellite overpasses and ground-based measurements, the allowed time difference between the satellite overpass and the spectroradiometer measurement was set to less than 5 minutes.

Recording frequencies of broadband and multichannel radiometers are listed in Table 4: averages were made over 1, 2, 5 or 10 minutes. The allowed time difference between the satellite overpass and the ground-based measurement was set to be less than half of the recording frequency. Eg. if ground-based data were recorded every minute, then the allowed time difference was set to less than 30 seconds. If the ground-based data was ten minute average, then the maximum time difference was set to be less than 5 minutes.

For the validation of overpass UV index both broadband and multichannel radiometers were used and the time difference between satellite overpass and ground-based data was limited to half of the recording frequency, as for the dose rate validation. All type of instruments (spectroradiometer, broadband and multichannel radiometer) were used for the validation of the erythemally weighted daily dose.

For all TROPOMI overpass pixels a ground-based measurement was chosen if found within the allowed time difference. No quality filtering was performed for the TROPOMI data. The distance between the TROPOMI pixel and the ground station was filtered to be less than 5 km, the SZA less than 80°, and following Tanskanen et al. (2007) the altitude difference between the altitude of the site and the TROPOMI pixel less than 500 m. For mountainous sites of Davos and Aosta the maximum distance between the TROPOMI pixel and the site was limited to be less than 3 km. The SZA of 80° was chosen to avoid very low UV irradiances, as for very low irradiances the ratio between satellite and ground-based data becomes unstable (Tanskanen et al., 2007). Also at SZAs smaller than 80° the effect of stray light in single monochromator Brewers (Bais et al., 1996) is avoided.

The relative difference $\rho$ between satellite data and ground-based data was calculated for each pair of satellite data (sat) and ground-based data (gr) using the following equation:

$$\rho = 100\% * [(\mathrm{sat} - \mathrm{gr})/\mathrm{gr}] \tag{1}$$

The median and 25th and 75th percentiles of the $\rho$ values were calculated for each site. The $W_{10}$ and $W_{20}$ from Tanskanen et al. (2007) were calculated also in this study. The $W_X$ is determined as percentage of satellite data which is within $X\%$ from ground-based data:

$$W_X = P(-X < \rho < X) \tag{2}$$

Similarly to Tanskanen et al. (2007) data sets were divided into subsets according to albedo. Snow cover was considered, when the albedo used by the TROPOMI UV processor (Kujanpää et al., 2020) was higher than 0.1, and the data set was divided into snow cover (SC) and snow free (SF) ground conditions. The albedo used by the TROPOMI UV processor is derived from albedo climatology (Kujanpää and Kalakoski, 2015), thus it may differ from the true albedo conditions of the site (Tanskanen et al., 2007; Bernhard et al., 2015). In addition, a subset of data, called "cloudfree" was selected. This subset includes data for which the cloud optical depth retrieved by the TROPOMI UV processor was lower than 0.5. This "cloudfree" data sets included both snow cover and snow free conditions. Here again, one needs to keep in mind, that it is the cloud optical depth as derived from the LUT of the TROPOMI UV processor, not the cloudiness observation from the site.

The spatial resolution of TROPOMI data is very high compared to older generation satellite instruments. This leads to a huge amount of data and at most sites several satellite pixels fulfilling the selection criteria were colocated with the same ground-based measurement. For example, at high latitudes, this increased the number of data with more than 5 pixels for each overpass. Thus, the sensitivity of the results was studied by comparing three different data selection methods for Villeneuve d'Ascq measurements: 1) Each TROPOMI pixel was treated as individual measurement, 2) the pixel nearest of the site was chosen, 3) the average of the TROPOMI pixels meeting the chosen limitations (time difference, SZA, altitude, distance) was used. The results did not differ significantly between the methods, and in this study the results were calculated for each pixel separately. Results are shown in the Fig. S13 and Table S6 of the Supplement material.

## 4 Results

Results for validation of overpass spectral irradiances, dose rates, UV index and daily doses are discussed separately in the following sections. Scatter plots, histograms and tables including the statistics were prepared for all studied UV parameters, and they are shown in the Supplement material of this paper. Here they are shown only for dose rate / UV index.

### 4.1 Spectral irradiances

TROPOMI overpass irradiances were compared with the following spectroradiometers listed in Table 3: SOD, HEL, VDA, DAV, DBR, AOS, OHP, ROM, THE, IZA, OPA, MEL and PAL. The statistics are shown in the Tables S1–S4 of the Supplement material. Scatter plots and histograms are showed in Figs. S1-S8. For irradiances at 305, 310, 324 and 380 nm the median $\rho$ was within $\pm10\%$ at 11, 9, 10 and 6 sites from the 13 sites (7 sites for 380 nm), respectively, for snow free ground conditions. For the four wavelengths, at all sites except one, more than 50% of satellite data were within $\pm20\%$ from ground measurements. During snow conditions the percentage of satellite data being within $\pm20\%$ from ground measurements decreased in Davos from more than 60% to around 20%. This is seen as a shift of $\rho$ towards negative values when comparing snow cover data set to snow free data set. For the other four sites which had data sets during snow cover, Sodankylä, Helsinki, Aosta and Palmer, no significant difference was observed between snow cover and snow free surfaces. At Palmer, a systematic underestimation of irradiances occured at all wavelengths. The median $\rho$ at 305 nm was -46% and -56% for snow free and snow covered surface, respectively. Satellite data had a positive bias at Davos, Aosta and Izaña, while at other sites the bias was more randomly distributed. The spread of the scatter plot was larger at 380 nm than at 305 nm, which is related to the influence of clouds: radiation distribution at short UV wavelengths is less affected by clouds than at longer wavelengths. For all stations, over 50% of cloudfree satellite data were within $\pm20\%$ from ground measurements.

### 4.2 Erythemally weighted dose rate and UV index

An example of TROPOMI overpass and ground-based erythemally weighted dose rate time series is shown in Fig. 2 together with the absolute difference for Uccle. TROPOMI data follow well daily variations in UV dose rates, and the absolute difference was less than 0.05 W/m$^2$. The validation results are shown by comparisons against spectroradiometers, broadband and multichannel radiometers in the following subsections.

#### 4.2.1 Validation against spectroradiometers

The comparison of TROPOMI overpass erythemally weighted dose rates against spectroradiometer measurements showed similar patterns as the comparison of single irradiances at 305 and 310 nm. The scatter plot and histograms are showed in Figs 3 and 4, respectively, and the statistics in Table 5. At ten and seven sites the median $\rho$ was within $\pm8\%$ and $\pm5\%$, respectively, for snow free conditions. As for irradiances, TROPOMI UV dose rates show a systematic negative bias at Palmer, with median $\rho$ of -45% during snow cover conditions. The histograms of $\rho$ are similar for snow cover and snow free conditions.

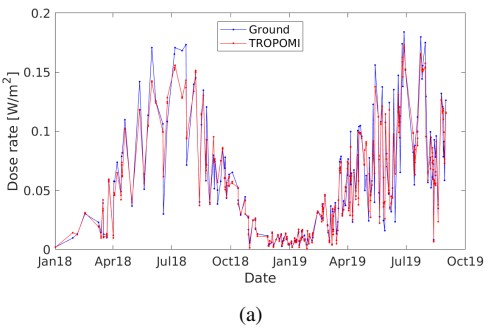 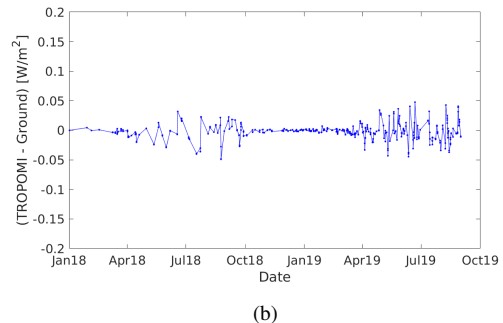

(a)                                                                              (b)

**Figure 2.** TROPOMI overpass and Brewer spectroradiometer a) erythemal dose rates, b) absolute difference of erythemal dose rates at Uccle, Belgium, during Jan 2018 – Aug 2019.

Also at the other sites which have data sets for snow cover conditions, there are no noticeable differences between snow cover and snow free conditions.

At all sites, except Palmer, over 60% of TROPOMI data are within ±20% from ground-based measurements. In Aosta and Izaña there is a large positive bias for some pixels. Large positive biases in TROPOMI UV data occur over these mountainous regions during cloudy conditions when the "rough terrain" quality flag is active and cloud optical depth is set to zero in the UV algorithm.

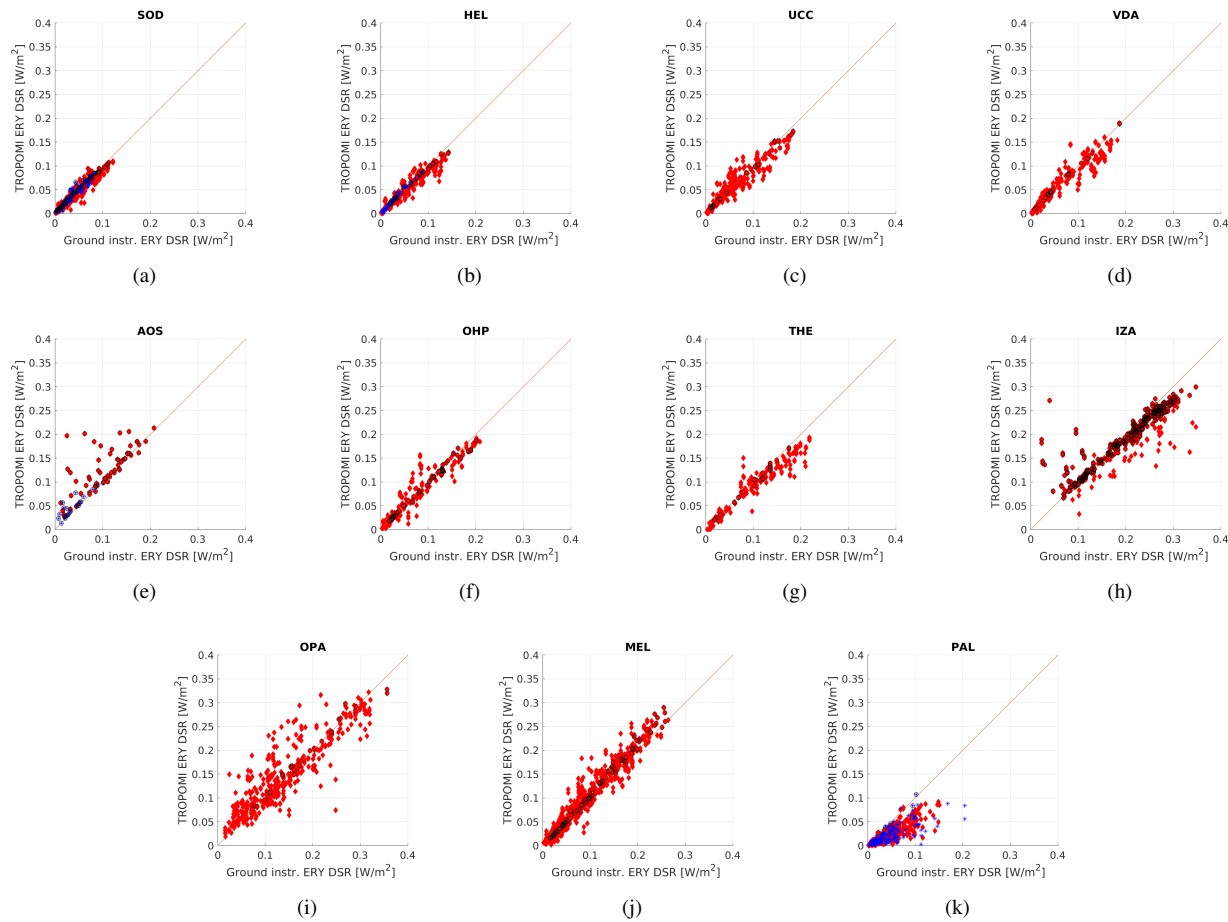

**Figure 3.** Erythemally weighted dose rates from spectroradiometer measurements and retrieved from satellite overpass at a) Sodankylä, b) Helsinki, c) Uccle, d) Villeneuve d'Ascq , e) Aosta, f) Haute-Provence, g) Thessaloniki, h) Izaña, i) Saint-Denis, j) Melbourne and k) Palmer. Red diamond denotes snow free surface, blue star snow cover and black circle cloudfree.

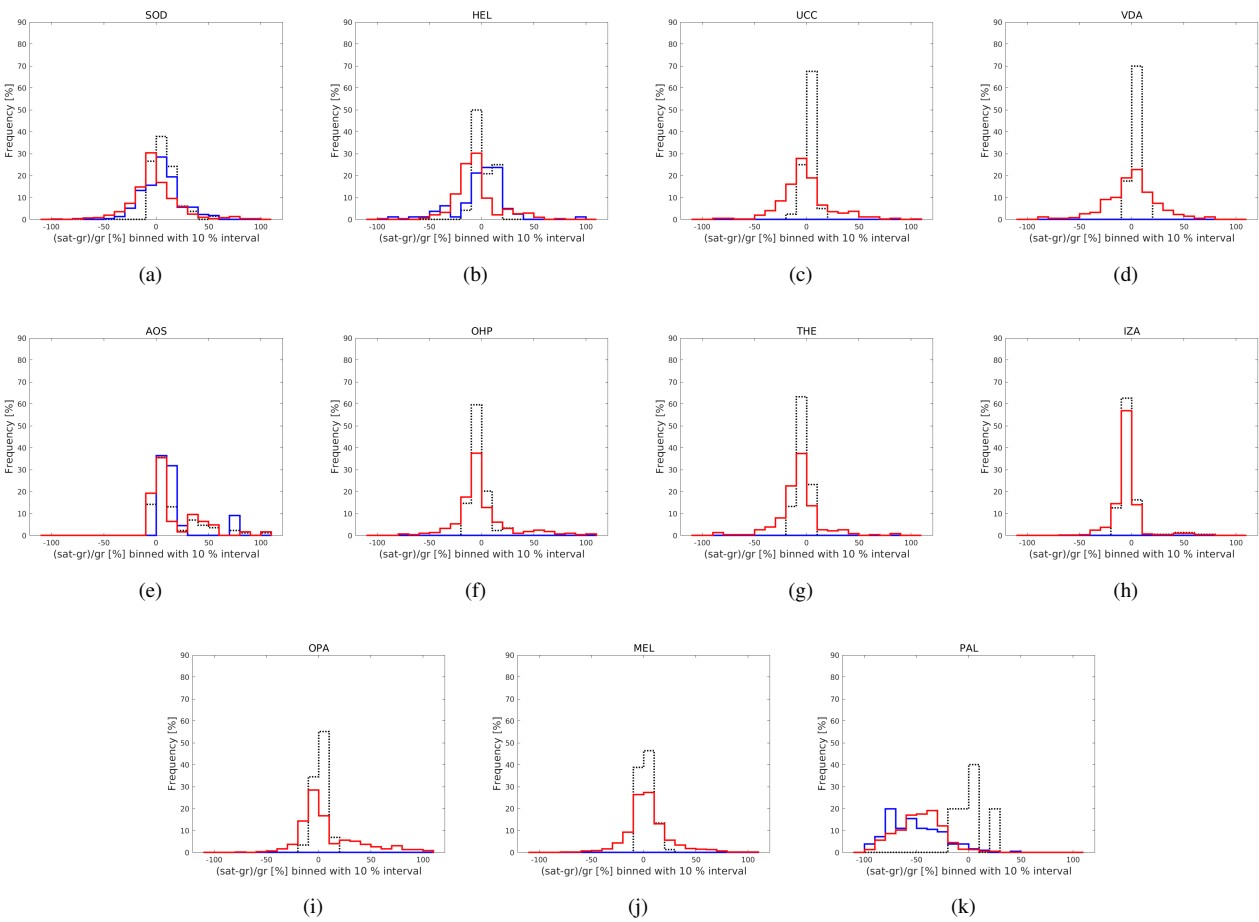

**Figure 4.** Histograms of relative difference between spectroradiometer measurements and satellite overpass erythemally weighted dose rates at a) Sodankylä, b) Helsinki, c) Uccle, d) Villeneuve d'Ascq , e) Aosta, f) Haute-Provence, g) Thessaloniki, h) Izaña, i) Saint-Denis, j) Melbourne and k) Palmer. Black dotted line denotes cloudfree, red snow free surface and blue snow cover on the ground.

**Table 5.** TROPOMI overpass erythemally weighted dose rates compared to spectroradiometer measurements. The percentage relative differences 100%*(sat-gr)/gr, their medians (median) and 25th (p25) and 75th (p75) percentiles were calculated. N is the number of measurement days included in the study. $W_{10}$ and $W_{20}$ are percentage of satellite data which are within 10% and 20% from ground measurements. CF=Cloudfree, SCAS=Snow cover at ground, all sky, SFAS=Snow free ground, all sky.

| Station | Conditions | N | median [%] | p25 [%] | p75 [%] | $W_{10}$ | $W_{20}$ |
|---|---|---|---|---|---|---|---|
| SOD | CF | 132 | 5.92 | -0.45 | 12.48 | 64.39 | 88.64 |
| SOD | SCAS | 211 | 4.09 | -6.49 | 14.50 | 44.08 | 76.78 |
| SOD | SFAS | 451 | -3.19 | -12.22 | 8.82 | 47.23 | 71.62 |
| HEL | CF | 48 | -0.86 | -4.24 | 8.80 | 70.83 | 100.00 |
| HEL | SCAS | 80 | 2.43 | -6.48 | 11.99 | 45.00 | 76.25 |
| HEL | SFAS | 275 | -8.11 | -15.22 | 0.29 | 40.00 | 67.64 |
| UCC | CF | 40 | 3.01 | -1.09 | 4.84 | 92.50 | 100.00 |
| UCC | SFAS | 399 | -4.42 | -13.89 | 6.64 | 46.87 | 69.42 |
| VDA | CF | 40 | 6.65 | 3.60 | 7.83 | 87.50 | 100.00 |
| VDA | SFAS | 337 | 1.63 | -13.17 | 14.14 | 41.84 | 64.39 |
| AOS | CF | 84 | 10.19 | 1.96 | 43.24 | 50.00 | 63.10 |
| AOS | SCAS | 22 | 15.00 | 6.68 | 73.36 | 36.36 | 68.18 |
| AOS | SFAS | 62 | 7.22 | 1.43 | 41.87 | 54.84 | 61.29 |
| OHP | CF | 89 | -2.44 | -7.19 | 0.25 | 79.78 | 96.63 |
| OHP | SFAS | 280 | -4.11 | -11.07 | 3.40 | 50.36 | 73.93 |
| THE | CF | 30 | -2.57 | -8.61 | -0.50 | 86.67 | 100.00 |
| THE | SFAS | 235 | -7.68 | -14.10 | -0.62 | 51.06 | 76.17 |
| IZA | CF | 386 | -4.50 | -7.34 | -0.11 | 79.02 | 91.97 |
| IZA | SFAS | 454 | -5.38 | -9.02 | -1.15 | 70.93 | 86.12 |
| OPA | CF | 29 | 1.29 | -1.18 | 6.32 | 89.66 | 100.00 |
| OPA | SFAS | 424 | 0.05 | -8.14 | 28.19 | 45.28 | 63.68 |
| MEL | CF | 142 | 2.34 | -2.02 | 7.56 | 85.21 | 98.59 |
| MEL | SFAS | 1074 | 2.24 | -6.36 | 12.34 | 53.91 | 76.16 |
| PAL | CF | 5 | 3.13 | -7.07 | 9.87 | 60.00 | 80.00 |
| PAL | SCAS | 180 | -56.64 | -72.29 | -34.24 | 5.56 | 10.56 |
| PAL | SFAS | 393 | -44.88 | -60.74 | -31.96 | 2.80 | 7.89 |

### 4.2.2 Validation against broadband and multiband radiometers

The scatter plots and histograms of TROPOMI overpass UV index and dose rates comparison against broadband and multi-channel radiometers are shown in Figs 5 and 6, respectively, and the statistics in Table 6. The number of colocated pixels is

much higher for broadband instruments than for spectroradiometers, as they measure continuously. At several sites (Jerusalem, Mahé, Antananarivo, Anse Quitor, Saint-Denis and Alice Springs), the UV index can be higher than 11, categorized as "extreme" UV (WMO, 1997). These extreme values are underestimated by TROPOMI, except in Alice Springs. The feature is pronounced in the Indian Ocean sites Mahé, Antananarivo and Anse Quitor. The strongest underestimation is seen in Mahé, where the median $\rho$ was -34% with the 25th and 75th percentiles of -40% and -20%, respectively.

For the other sites, the median $\rho$ for snow free conditions was between -1 and -10%. At the high latitude site of Ny-Ålesund where snow covers the surface almost half a year, and at the mountainous site of Finse, similar underestimation to Palmer is seen. However, at Ny-Ålesund and Finse, differences occur between snow cover and snow free data sets. The medians $\rho$ for snow free conditions are -10% at both sites, and for snow cover conditions -30% and -65% at Ny-Ålesund and Finse, respectively. The difference of snow cover and snow free conditions is distinctly seen in the histogram (Fig. 6a and 6c). Also at Davos and Andøya, underestimation occurred during snow cover, and $\rho$ differed between snow cover and snow free data sets. The median $\rho$ was approximately -5% and -35% for snow free and snow covered conditions, respectively, at both sites. At Blindern, the same feature was seen, but with a smaller difference between the two conditions: -5% and -20% for snow free and snow cover conditions, respectively. At Marambio, the Antarctic station which has snow cover all year round, the underestimation was similar to Blindern with median $\rho$ of -20%.

The median $\rho$ for cloudfree conditions was within $\pm10\%$ for all sites except the Indian Ocean sites, Mahé, Antananarivo and Anse Quitor, and the Israeli site of Jerusalem. At 8 sites the median was within $\pm5\%$ for cloudfree conditions.

The effect of taking into account quality flags was evaluated for the site of Davos. Data for which the quality value number UVQAV was less than 0.5 were excluded (see Section 2.1 for explanation of UVQAV). This removed e.g., unreliable values when the cloud optical depth was set to 0 due to the flagging. Indeed, as mentioned in Section 2.2, Davos is a mountainous site with heterogeneous albedo during the winter. Setting a limit of 0.5 for the UVQAV, results in removing satellite observations with at least two of the following warnings: "rough_terrain", "alb_hetero" or "clearsky_assumed". This procedure reduced the number of data points by about half, and removed most data points where satellite estimates exceed the ground measurements. This resulted in a shift of median relative differences towards more negative values: From -24% to -57% and from -6% to -13 % for snow cover and snow free conditions, respectively. The statistics and scatter plots of the study are shown in Supplement material Table S7 and Fig. S14.

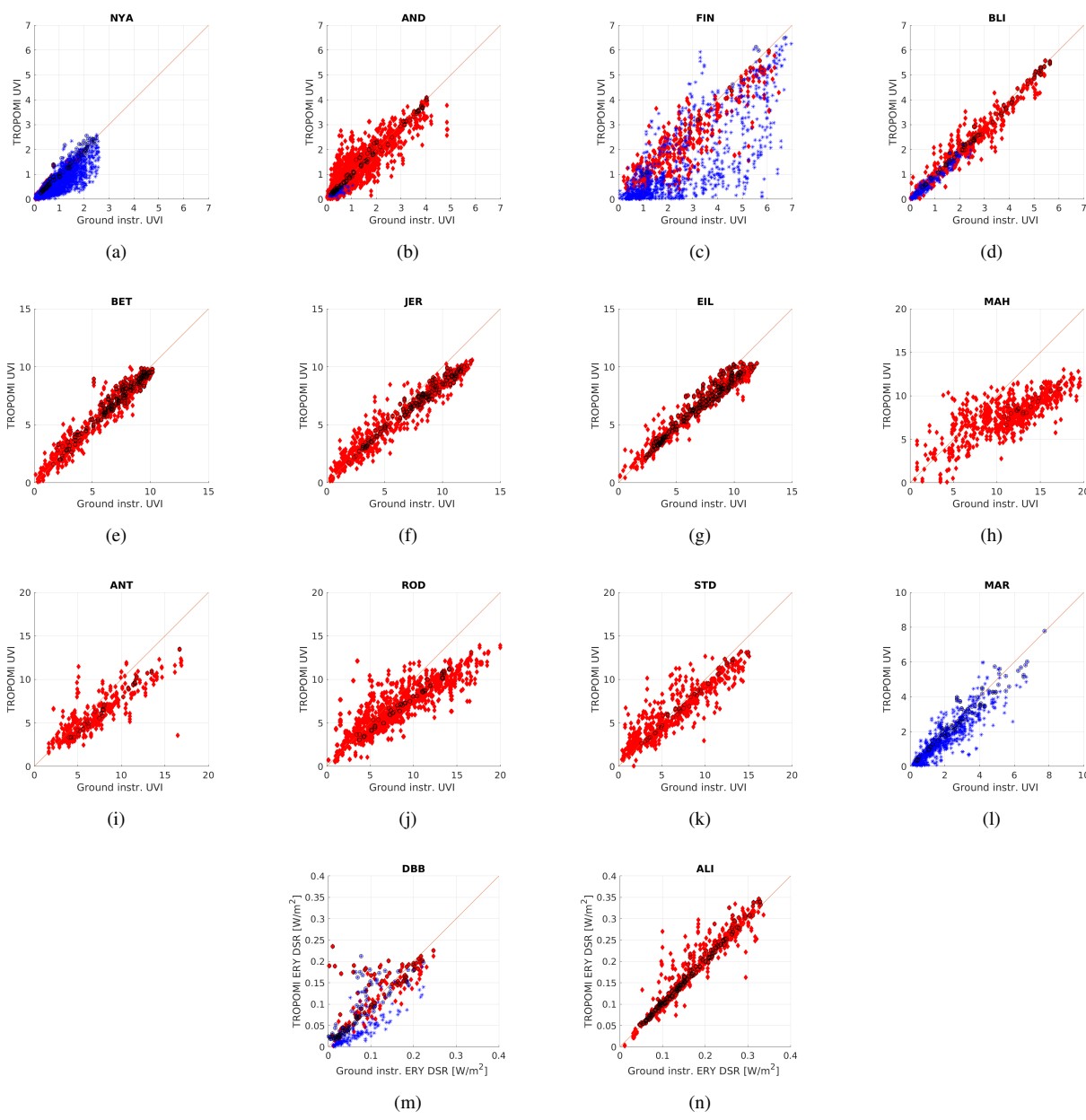

**Figure 5.** UV index from radiometer measurements and retrieved from satellite overpass at a) Ny-Ålesund, b) Andøya, c) Finse, d) Blindern, e) Bet Dagan, f) Jerusalem, g) Eilat, h) Mahé, i) Antananarivo, j) Anse Quitor, k) Saint-Denis, l) Marambio and erythemally weighted dose rates at m) Davos and n) Alice Springs. Red diamond denotes snow free surface, blue star snow cover and black circle cloudfree.

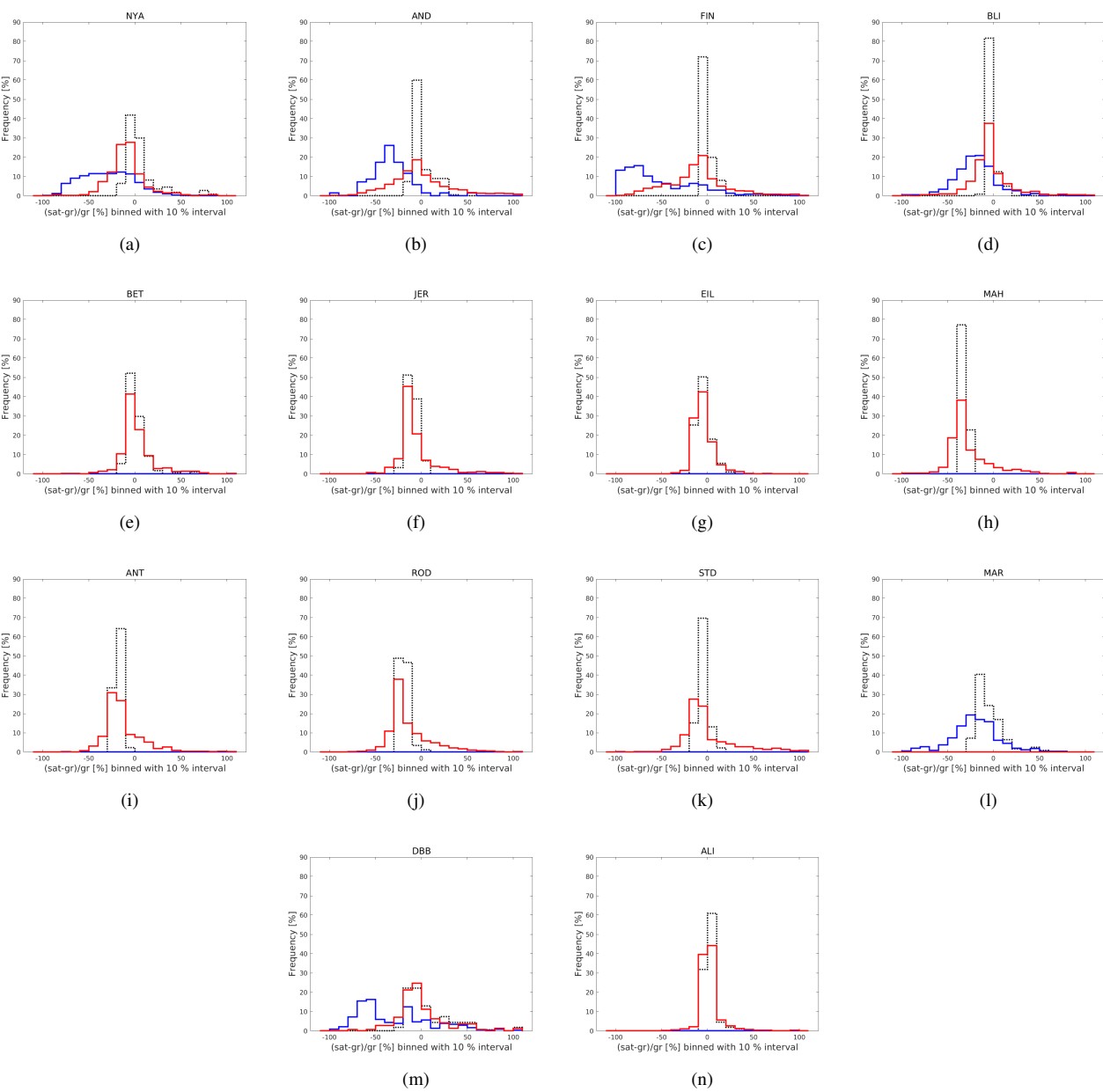

**Figure 6.** Histograms of relative difference between radiometer measurements and satellite overpass UV index at a) Ny-Ålesund, b) Andøya, c) Finse, d) Blindern, e) Bet Dagan, f) Jerusalem, g) Eilat, h) Mahé, i) Antananarivo, j) Anse Quitor, k) Saint-Denis, l) Marambio and overpass erythemally weighted dose rates at m) Davos and n) Alice Springs. Black dotted line denotes cloudfree, red snow free surface and blue snow cover on the ground.

**Table 6.** TROPOMI overpass UV index/erythemally weighted dose rates compared to broadband and multichannel radiometer measurements. The percentage relative differences 100%*(sat-gr)/gr, their medians (median) and 25th (p25) and 75th (p75) percentiles were calculated. N is the number of measurement days included in the study. $W_{10}$ and $W_{20}$ are percentage of satellite data which are within 10% and 20% from ground measurements. CF=Cloudfree, SCAS=Snow cover at ground, all sky, SFAS=Snow free ground, all sky.

| Station | Conditions | N | median [%] | p25 [%] | p75 [%] | $W_{10}$ | $W_{20}$ |
|---------|-----------|------|-----------|---------|---------|--------|--------|
| NYA | CF | 110 | 0.64 | -4.18 | 8.55 | 71.82 | 86.36 |
| NYA | SCAS | 3333 | -30.28 | -52.24 | -9.25 | 18.33 | 34.26 |
| NYA | SFAS | 832 | -9.96 | -18.93 | -1.49 | 38.94 | 70.07 |
| AND | CF | 177 | -2.71 | -5.53 | 1.48 | 73.45 | 89.83 |
| AND | SCAS | 69 | -33.66 | -43.95 | -22.00 | 7.25 | 18.84 |
| AND | SFAS | 2378 | -3.23 | -18.94 | 22.13 | 29.48 | 49.54 |
| FIN | CF | 25 | -2.17 | -4.52 | 0.19 | 92.00 | 100.00 |
| FIN | SCAS | 1099 | -64.96 | -82.27 | -21.73 | 8.55 | 17.93 |
| FIN | SFAS | 446 | -9.86 | -28.60 | 3.79 | 29.60 | 50.67 |
| BLI | CF | 120 | -3.76 | -5.84 | -1.11 | 94.17 | 100.00 |
| BLI | SCAS | 345 | -20.00 | -32.41 | -8.06 | 20.87 | 44.93 |
| BLI | SFAS | 620 | -5.26 | -14.33 | 2.77 | 49.03 | 74.03 |
| DBB | CF | 163 | 1.90 | -9.26 | 39.66 | 34.97 | 61.35 |
| DBB | SCAS | 233 | -37.33 | -60.86 | 2.90 | 10.30 | 24.03 |
| DBB | SFAS | 142 | -4.88 | -12.92 | 16.17 | 35.92 | 63.38 |
| BET | CF | 305 | -0.85 | -4.06 | 4.13 | 81.97 | 96.39 |
| BET | SFAS | 886 | -1.21 | -6.28 | 6.76 | 64.33 | 84.09 |
| JER | CF | 250 | -10.82 | -14.94 | -6.86 | 45.60 | 96.80 |
| JER | SFAS | 899 | -12.10 | -16.83 | -2.98 | 27.81 | 77.09 |
| EIL | CF | 406 | -5.31 | -10.10 | -0.25 | 68.23 | 99.01 |
| EIL | SFAS | 872 | -5.99 | -11.48 | 0.63 | 59.06 | 92.66 |
| MAH | CF | 22 | -32.84 | -36.91 | -31.24 | 0.00 | 0.00 |
| MAH | SFAS | 719 | -33.81 | -39.83 | -19.79 | 8.48 | 17.80 |
| ANT | CF | 42 | -18.16 | -21.70 | -16.43 | 2.38 | 66.67 |
| ANT | SFAS | 431 | -17.81 | -25.34 | -5.12 | 16.71 | 48.49 |
| ROD | CF | 88 | -19.69 | -21.81 | -17.78 | 4.55 | 51.14 |
| ROD | SFAS | 1527 | -20.82 | -26.54 | -3.69 | 15.46 | 35.30 |
| STD | CF | 46 | -6.69 | -9.01 | -4.31 | 82.61 | 100.00 |
| STD | SFAS | 610 | -8.11 | -14.96 | 14.35 | 30.33 | 63.11 |
| ALI | CF | 286 | 1.79 | -0.72 | 5.59 | 92.66 | 97.20 |
| ALI | SFAS | 898 | 0.75 | -2.62 | 5.33 | 83.74 | 91.54 |
| MAR | CF | 124 | -9.68 | -14.84 | 1.85 | 41.13 | 87.90 |
| MAR | SCAS | 1169 | -20.00 | -33.53 | -4.76 | 21.90 | 43.37 |

## 4.3 Erythemally weighted daily dose

TROPOMI erythemally weighted daily doses were compared against daily doses derived from spectroradiometer measurements (SOD, HEL, UCC, VDA, AOS, OHP, IZA, OPA, MEL and PAL), multichannel radiometers (NYA, AND, FIN, BLI and MAR) and a broadband radiometer (ALI). The scatter plots, histograms and statistics are showed in the Figs. S9-S10, S11-S12 and Table S5 of the Supplement material, respectively.

As the satellite daily dose is calculated using the assumption that the cloudiness retrieved by the satellite during the overpass would be the same during the whole day, which is an oversimplified assumption for many sites, larger deviation in the results were expected than for overpass dose rates. At all sites except Palmer, Aosta, Finse and Ny-Ålesund, the increased deviation was seen in both positive and negative biases. However, the median $\rho$ was within $\pm10\%$ and $\pm5\%$ at 11 and 8 sites, respectively. The total number of sites providing daily doses for this study was 16. At all sites, except Palmer, over 50% of satellite data were within $\pm20\%$ from ground-based measurements during snow free surface conditions. Marambio was always snow-covered.

At Palmer, the pattern was similar to results of the dose rate validation indicating large underestimation with median $\rho$ of -49% and -62% for snow free and snow covered conditions. Also at Ny-Ålesund there was significant underestimation, but daily doses differed from dose rates by having also several overestimation cases. At Ny-Ålesund the median $\rho$ was -12% and -33% for snow free and snow cover conditions, while during cloudfree conditions there was an overestimation with a median $\rho$ of 6%. This could occur at situations, when the sky was cloudless during the overpass, but later changed towards more cloudy conditions, or would have been more cloudy before the overpass. Also at Andøya, Blindern and Finse large differences between the median of snow free ($\rho$=-2%, $\rho$=-5% and $\rho$=-12%, respectively) and snow cover ($\rho$=-36%, $\rho$=-22% and $\rho$=-66%, respectively) condition occurred. At Sodankylä and Helsinki the difference between the median of snow free and snow cover conditions was less than 10% and 4%, respectively.

At Aosta and Izaña there were large cloudfree overestimations. At Aosta, the reason is the non-homogeneous topography around this mountainous site. At Aosta almost all satellite data is flagged with the "rough_terrain" and "clearsky_assumed" flags, and the data agree with measurements only during real clear sky conditions. At Izaña there were similar overestimation cases but also several underestimations. The underestimations can be due to situations, when the station is above the clouds, but the satellite interprets surrounding clouds, which in reality are below the station, as cloudiness of the site.

## 5 Discussion

Tanskanen et al. (2007) summarized the key validation statistics of OMI daily dose by plotting $W_{20}$ as a function of median $\rho$ (Fig. 6 in Tanskanen et al. (2007)). The same was done for this validation. Results for the cloudfree datasets, with the cloud optical depth input parameter of the TROPOMI UV algorithm lower than 0.5, were also included in the plot. Cloudfree criteria is not reflecting actual cloudiness conditions, but the cloud optical depth retrieved from the first LUT in the TROPOMI UV algorithm. If a perfect agreement between satellite and ground-based data is found, the surface albedo climatology and aerosol climatology are most probably representative for the actual surface albedo and aerosol conditions, respectively.

Figure 7 shows results for overpass irradiance validation, and Fig. 8 for overpass dose rates and daily doses. One notable difference between the OMI results from Tanskanen et al. (2007) and the results of the TROPOMI validation study is that positive bias due to tropospheric extinction is missing from TROPOMI results. That is due to the correction for absorbing aerosols which was not implemented in the OMI data in the study of Tanskanen et al. (2007). The current OMI UV algorithm is updated with the absorbing aerosol correction method described in Arola et al. (2009) and the same method is used in 470 the TROPOMI UV algorithm. Thessaloniki is a site for which aerosols are an important factor affecting UV radiation (e.g., Fountoulakis et al., 2016a). Tanskanen et al. (2007) found a median difference of 16% (OMI/ground -1) between OMI and ground based erythemally weighted daily dose at Thessaloniki, while the TROPOMI validation showed an underestimation of -8% with 19% more satellite retrievals within ±20% from ground-based dose rate measurements. Even if no need for improvement was detected in this specific study, actual aerosol data from e.g. satellite retrievals would be a good improvement 475 for taking into account local aerosol anomalies.

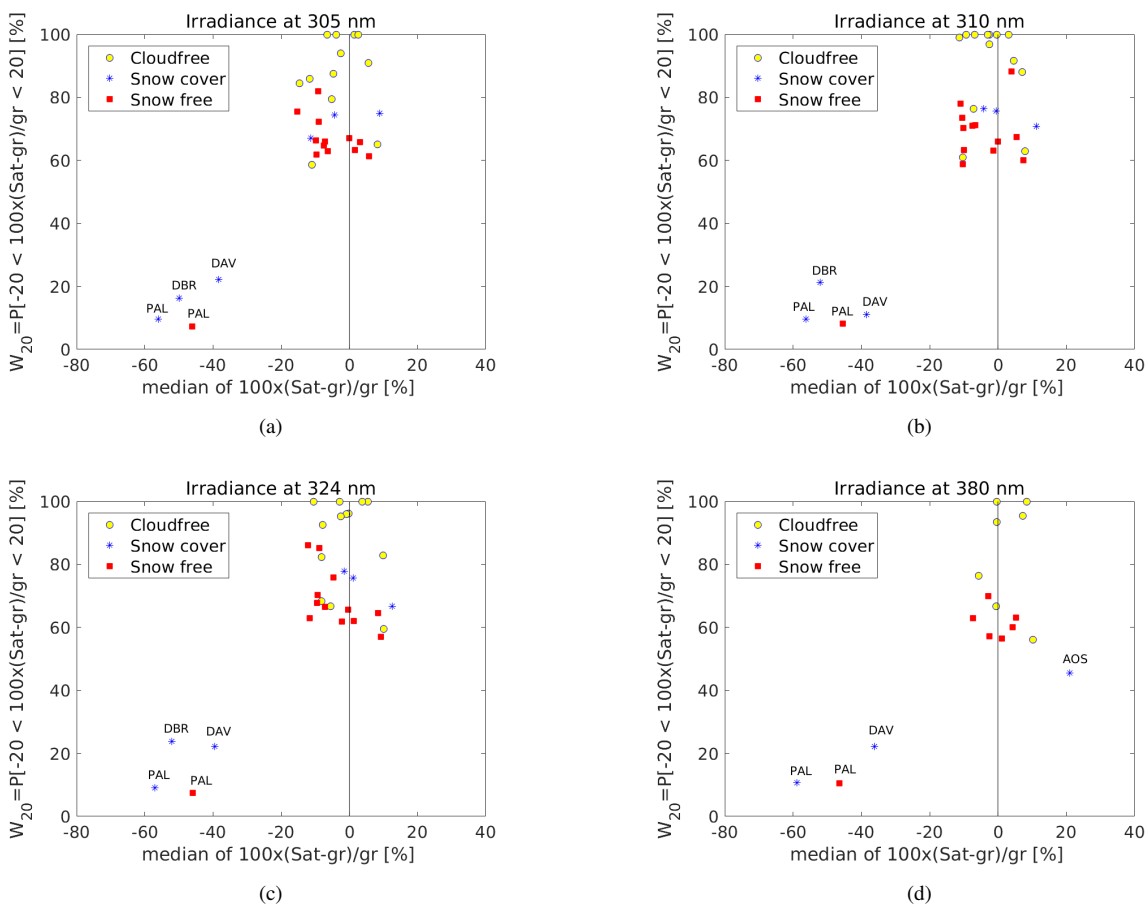

**Figure 7.** $W_{20}$ in function of $\rho$ for validation sites measuring overpass irradiance at a) 305, b) 310, c) 324 and d) 380 nm.

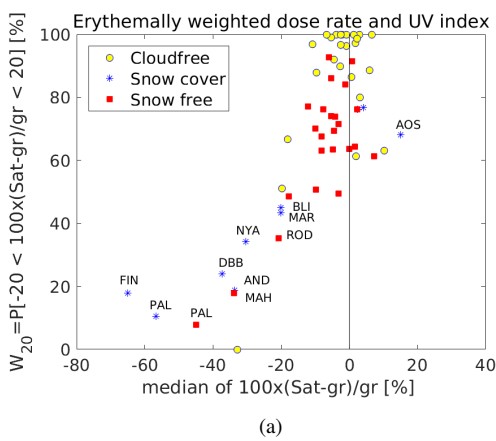 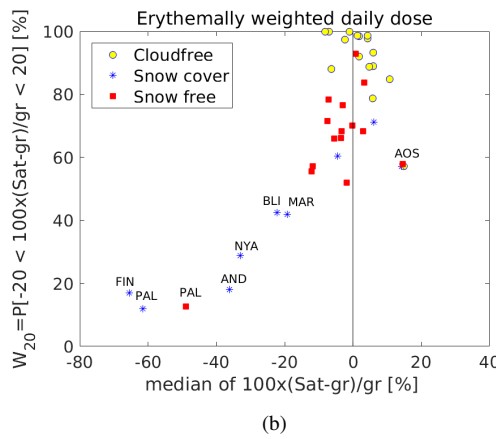

**Figure 8.** $W_{20}$ in function of $\rho$ for a) overpass erythemally weighted dose rate and UV index b) erythemally weighted daily dose.

Kalliskota et al. (2000) found an underestimation of TOMS UV daily dose at Ushuaia, Argentina, and Palmer, and an overestimation in San Diego, U.S. Also TROPOMI underestimated the daily dose at Palmer (median $\rho$ of -49%) for snow free surface, while agreed quite well (median $\rho$ of -8%) during cloudfree conditions. The monthly average underestimation of erythemally weighted daily dose was -35% for TOMS and the median underestimation (OMI/ground-1) was -33% for
OMI. The results of OMI refer to those calculated using the cloud correction method based on the plane-parallel cloud model (Tanskanen et al., 2007). OMI overestimated erythemally weighted daily doses by 10% (median of (OMI/ground -1)) for snow covered conditions at Sodankylä. For TROPOMI, the corresponding overestimation was 6%. For snow free conditions at Sodankylä, the median differences between satellite retrievals and ground-based measurements were 6% and -3% for OMI and TROPOMI, respectively.

Bernhard et al. (2015) studied in detail comparison of the OMI UV data against ground-based measurements at high latitudes and focused on the albedo effect. The sites of Ny-Ålesund, Sodankylä, Finse and Blindern were also included in their study. For Sodankylä, the results agreed with those of Tanskanen et al. (2007) (median $\rho$ = 11% for snow free and 6% for snow covered conditions). At Ny-Ålesund, Finse and Blindern the median differences of erythemally weighted daily dose (OMI/ground - 1) were -42%, -53% and -6%, respectively, for snow covered conditions, and 6%, 1% and 7%, respectively, for snow free
conditions. The corresponding values from the TROPOMI validation were -33%, -66% and -22%, respectively, for snow covered conditions, and -12%, -12% and -6% for snow free conditions. The results from Bernhard et al. (2015) were calculated from two months of data (one month in spring when the ground was snow covered and one month in summer when it was snow free), while the TROPOMI validation results were calculated from the data set covering the whole year.

As discussed in Tanskanen et al. (2007), the OMI UV underestimation in Palmer is mostly due to unreasonably small
values of surface albedo used by the OMI UV processor which results in an overestimate of the cloud optical thickness by misinterpreting reflections from the surface. This is also the dominant reason why TROPOMI underestimates UV radiation at Palmer and other sites with challenging snow albedo conditions (e.g., Ny-Ålesund, Finse, Davos). The albedo used by

TROPOMI was compared with those calculated from SUV-100 spectra at Palmer. The SUV-100 albedo ranged from 0.2 to 0.7 while the TROPOMI albedo ranged from 0.05 to 0.4. The largest differences between the two albedo data sets was around 0.3.

Surface albedo in Antarctica has a strong impact on UV radiation by increasing reflections from the surface and by multiple scattering between surface, clouds and atmosphere, which reduce the attenuation by clouds. According to Nichol et al. (2003) "a cloud with an optical depth of 10 reduces the UV irradiance, relative to clear-sky conditions, by 40% when the surface albedo is 0.05 as compared with reductions of 20% and 10% for surface albedos of 0.80 and 0.96, respectively."

At sites with homogeneous albedo conditions, even at locations with high surface albedo like Sodankylä, the differences
between satellite retrievals and ground-based data are much smaller than for non-homogeneous conditions. In this study even the two Antarctic sites, Palmer and Marambio, differed from each other regarding the underestimation during snow cover conditions. At Marambio, the median $\rho$ of erythemally weighted dose rate and daily doses was -20%, while at Palmer it was -62% for snow covered conditions. The smaller bias at Marambio can be attributed to the fact that the albedo values used in the TROPOMI UV algorithm (ranging from 0.2 to 0.7) are more realistic than at Palmer.

At high latitudes also long-term changes in the effective albedo resulting from climate change have to be considered in the future. Already now, at some Arctic sites the length of snow cover period has shortened by several weeks compared to a couple of decades ago (Bernhard, 2011; Luomaranta et al., 2019; Takala et al., 2011). That might be one reason for the positive median $\rho$ of Sodankylä and Helsinki dose rates compared to snow free negative median $\rho$: If the climatological albedo used by TROPOMI is too high it can lead to underestimation of cloudiness. The results of Bernhard et al. (2015) showed that for the
OMI instrument, overestimations and underestimations of up to 55% and -59%, respectively, were due to errors in the albedo climatology used in the OMI UV algorithm. In the ideal case, the surface albedo input would have the same space resolution as TROPOMI and would follow actual albedo changes in time.

As discussed also by Tanskanen et al. (2007) validation results become unstable when UV irradiance is low, which is the case when SZAs or cloud optical depths are very large. Then, even small absolute differences are seen as large relative differences.
This is frequent at high latitudes where large SZAs are present even at noon (overpass time) in winters. An example is shown in Fig. 9 for Ny-Ålesund, a site where UV index underestimations of more than -50% were found. Most of differences in the UV index are less than 0.5 when the whole data set is studied (including snow covered and snow free conditions) (Fig. 9a). The maximum differences are 1.5 for the lowest SZAs during snow cover conditions (9b).

For many sites with uniform topography, stable albedo conditions, or predictable changes in albedo over the course of a
525 year, TROPOMI UV data products agreed with ground-based measurements to within ±5%. Aosta is a good example of problems faced when retrieving satellite UV for mountainous sites. The non-homogeneous topography leads to uncertainties in the retrieval of cloud optical depth, which often result in a cloud optical depth of zero when in fact clouds were present, which in turn leads to overestimations of UV radiation by TROPOMI. Quality flags related to topography and cloud optical thickness are included in the TROPOMI surface UV product and could be used to identify such cases of heterogeneous topography.

In this study, the cloud optical depth was forced to zero when the UV product quality flag showed rough terrain. Following previous experience this has worked for mountain sites, e.g. Tibet region, where the site is most of the time above the clouds. However, this study showed that there are big challenges e.g., in the Alps, where the topography is strongly non-homogeneous

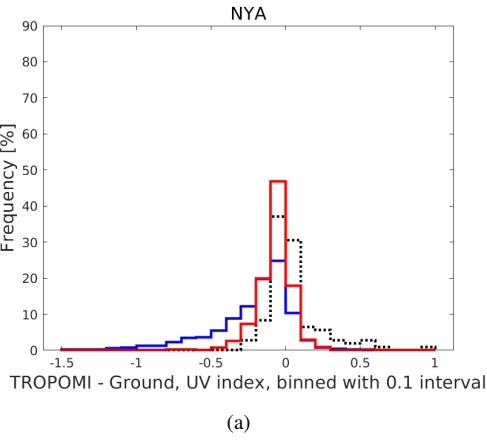 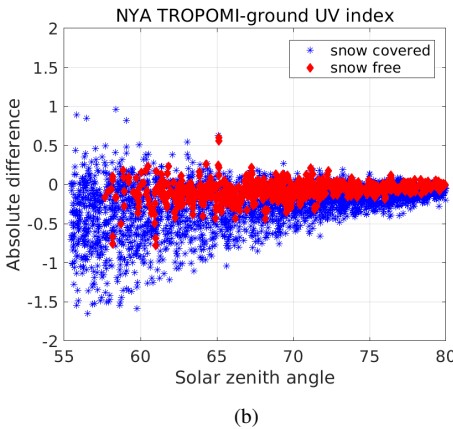

(a)                                            (b)

**Figure 9.** The absolute difference between the TROPOMI overpass UV index and ground-based measurement at Ny-Ålesund as a) histogram binned with 0.1 interval and b) a function of SZA.

and the site is located in the valley, e.g., Aosta and Davos. The satellite pixel, which is around 7x4 km$^2$ can include in such mountainous area high elevation differences, with one part of the pixel being inside the cloud and the other one outside the cloud. However, estimating UV radiation from space at locations with non-homogeneous terrain will always be challenging.

As the TROPOMI pixel is between 7x3.5 km$^2$ (nadir) and 9x14 km$^2$ (edge of swath), also the surroundings of a site are included in the pixel. Coastal high-latitude sites are particularly challenging (e.g., NYA, AND, PAL and MAR) because both open ocean and land covered by snow can be within the area of a pixel. The albedo is determined by the climatology of the central point of the pixel which can be open ocean even if the radiation field of the site is characterized by the surrounding snow cover. As for heterogeneous topography, the quality flagging also includes a flag for heterogeneous surface albedo.

The same applies for the impact of cloudiness when clouds are non-uniformly located around the site due to topography or changes in surface (e.g. sea/ground). For example the site itself is free from clouds but there can be small cumulus clouds at the edge of the pixel which increase the reflection towards the satellite. In that case TROPOMI would most probably underestimate the UV irradiance as the small fraction of clouds is considered as cloudiness of the whole pixel. Under scattered clouds, the UV radiation at the surface can be larger than during clear skies (Calbó et al., 2005; Jégou et al., 2011). This phenomenon occurs when the direct radiation from the Sun is not obstructed and additional radiation is scattered by clouds to the radiometer at the surface. The TROPOMI algorithm does not consider these situations, resulting UV levels that are too low. This phenomenon is likely one reason for the underestimation found in the TROPOMI UV dose rates during high UV levels at tropical sites.

The calculation of daily UV doses is based on the cloud optical depth at the time of the satellite overpass and the assumption that cloud cover is constant throughout this day. While this simplification leads to uncertainties, day-to-day variations in daily doses measured by the radiometers at the ground are generally well reproduced by TROPOMI. One could expect systematic biases at sites in which a diurnal cycle occurs, e.g. differences in cloudiness between morning and afternoon like orographic

clouds forming in the afternoon. This is the case for Saint-Denis, however, the effect on the bias of TROPOMI data is not very large. Rapidly changing cloudiness can be seen as large deviation between UV irradiances measured at successive pixels.

## 5.1 Comparison with other satellite surface UV products

TROPOMI is planned to continue OMI surface UV time series. A detailed comparison of OMI and TROPOMI surface UV product is needed and it will be subject for future study. Many publications have discussed OMI surface UV products, but only few included the same sites and the same UV parameters. In Tables 7–8 TROPOMI surface UV product validation results are shown together with those from OMI and GOME-2 studies at sites having comparable results. When interpreting the results, one should keep in mind, that each study has different data time periods, spatial and time difference, quality criteria, and the overpass time of the day varies between satellites. In addition, the pixel size of each satellite instrument is different. These differences suggest that validation results depend on actual cloudiness and other atmospheric conditions like aerosols at polluted sites or sites affected by seasonal aerosol or dust events, as well as surface albedo conditions.

In Table 7 GOME-2 stands for the EUMETSAT Surface UV Data Record 2007-2017 generated in the framework of the Satellite Application Facility on Atmospheric Composition Monitoring (AC SAF) (Kujanpää, 2018). It is a multimission product which is produced using as input total ozone column from GOME-2/Metop-A and/or GOME-2/Metop-B, and cloud optical depth from AVHRR-3 onboard Metop-A, Metop-B, NOAA-18 and NOAA-19. The main improvement of the data record over the operational OUV product is that one uniform algorithm version is used for the whole time period, and that climatological aerosol optical thickness and surface UV albedo inputs are changed from climatological values to actual daily values. The effect of using surface albedo values which corresponds better to actual conditions at the site is seen for Palmer (Table 7), where the median relative difference between GOME-2 data and ground-based measurements is 6%, while for TROPOMI and OMI data it is -49% and -33%, respectively, for snow free conditions. Even if Table 8 shows different statistics for 324 nm irradiance validation of TROPOMI (median) and OMI (mean), it can be concluded that differences between TROPOMI and ground-based measurements are smaller than the ones between OMI and ground-based measurements. A possible reason is the smaller pixel size of TROPOMI, which better corresponds to the field of view of ground instruments. A noticeable difference is also that TROPOMI underestimates while OMI overestimates irradiances compared to ground data.

Fioletov et al. (2002) showed that TOMS overestimated surface UV on average by 9-10%, Tanskanen et al. (2007) found OMI to have median overestimations between 0-10% and this study shows that TROPOMI median relative differences to ground-based measurements are within $\pm 5\%$ at several sites even if TROPOMI surface UV tends to be lower than ground-based data. The smaller pixel size of TROPOMI compared to OMI suggest that validation results of TROPOMI are more representative of ground-based measurement conditions, e.g., regarding cloudiness. Summarized TROPOMI and OMI validation results are of the same magnitude (within $\pm 10\%$ for sites with homogeneous conditions), but OMI usually overestimates while TROPOMI underestimates. Further analyses are needed to detect the effects of, e.g., differences in radiative transfer models and the way they take into account cloudiness. Studies should be done with spatially and temporally corresponding data sets.

**Table 7.** OMI (satellite/ground-1) and GOME-2 erythemally weighted daily dose validation results together with TROPOMI validation results. The median of relative difference (%) is shown. The values in parenthesis are for snow-cover conditions. In GOME-2 results for snow-free and snow-cover conditions are not calculated separately.

| Site | TROPOMI | OMI Tanskanen et al. (2007) | OMI Bernhard et al. (2015) | GOME-2 Lakkala et al. (2019) |
|------|---------|------------------------------|-----------------------------|-------------------------------|
| NYA | -12 (-33) | | 6 (-42) | |
| AND | -2 (-36) | | 17 (-4) | |
| SOD | -3 (6) | 6 (10) | 6 (11) | -2 |
| FIN | -12 (-66) | | 1 (-53) | |
| HEL | -7 (-4) | | | 5 |
| BLI | -6 (-22) | | 7 (-6) | |
| UCC | -0.2 | | | 8 |
| AOS | 15 (14) | | | -12 |
| PAL | -49 (-62) | -33 (-63) | | 6 |

**Table 8.** OMI irradiance at 324 nm validation results (satellite/ground-1) by Arola et al. (2009) together with TROPOMI validation results. For THE, OMI overpass erythemal dose rate comparison results by Zempila et al. (2018) are also shown.

| Site | TROPOMI median (%) | OMI |
|------|--------------------|-----|
| VDA irr. at 324 nm | -0.4 | 14 (mean %) |
| ROM irr. at 324 nm | -12 | 23 (mean %) |
| THE irr. at 324 nm | -9 | 16 (mean %) |
| THE eryth. dose rate | -8 | 5.1 (median %) |

## 6 Conclusions

The TROPOMI Surface UV Radiation Product was validated against ground-based measurements at 25 sites for the period from 1 Jan 2018 to 31 Aug 2019. TROPOMI overpass irradiances at 305, 310, 324 and 380 nm were compared against spectroradiometer measurements from 13 sites (7 sites for 380 nm). No major differences between results of different wavelengths were found, except that cloudiness affected the irradiance at 380 nm by enlarging the spread of the deviation from ground-based measurements. The median relative difference between TROPOMI data and ground-based measurements was within $\pm 10\%$ at 11, 9, 10 and 6 sites for irradiance at 305, 310, 324 and 380 nm, respectively, during snow free surface conditions. More than half of the satellite data were within $\pm 20\%$ from ground-based measurements.

TROPOMI overpass erythemally weighted dose rates were compared against dose rates measured by spectroradiometers at 11 sites and by broadband or multiband radiometers at two sites. The TROPOMI overpass UV index, which is directly proportional to the erythemally weighted dose rate, was compared against broadband and multichannel radiometer measurements at 12 sites. These validation results showed that the median relative difference between TROPOMI and ground-based dose rates was within $\pm 10\%$ and $\pm 5\%$ at 18 and 10 sites, respectively, for snow free surface conditions.

TROPOMI erythemally weighted daily doses were compared against spectroradiometer, broadband and multichannel radiometer measurements at 16 sites. The median relative difference was within $\pm 10\%$ and $\pm 5\%$ at 11 and 8 sites, respectively, for snow free surface conditions. For both dose rates and daily doses, $60 - 80\%$ of TROPOMI data were within $\pm 20\%$ from ground-based data at most of the sites for snow free surface conditions.

For all UV parameters an increased over- or underestimation was found at challenging conditions for satellite retrievals. Those are related to non-homogeneous topography (large altitude differences within the satellite pixel), non-homogeneous surface albedo (e.g. open water + snow cover on land within the satellite pixel), surface albedo differing from the albedo climatology used by the TROPOMI (at high latitudes where year-to-year changes are large) and high UV levels in tropics. The TROPOMI UV parameters include quality flagging to help identifying many of the above mentioned cases.

Retrieving the correct cloud optical depth is still challenging over snow albedo when reflections from snow and clouds are misinterpreted or confused with each other. The underestimation in satellite retrievals is related to the albedo climatology used by TROPOMI, which should be updated. Also, the current albedo climatology does not change from year to year while the actual albedo (e.g., timing of snow melt) can change a lot. The challenge is that the albedo climatology should be retrieved in at least as small pixel size as used by the TROPOMI, preferably from the TROPOMI data itself.

The TROPOMI Surface UV Radiation Product continues the former TOMS and OMI UV time series with an upgraded spatial resolution. A preliminary comparison with earlier satellite instrument (TOMS, OMI and GOME-2) validation results shows that all agree within the same magnitude ($\pm 10\%$) with ground-based measurements at sites with homogeneous conditions. The nominal life time of TROPOMI is 7 years after which the S5P mission will hopefully be extended as has been the case for e.g. OMI with already over 15 years of observations. Together these UV time series based on satellite retrievals form a unique 30 years long global data set which can be used for multiple UV impact studies all over the World. For this purpose special efforts to develop homogenized long term satellite time series are needed.

*Data availability.* The station overpass files and documentation are available from https://nsdc.fmi.fi/data/data_s5puv.php. Ground-based data are available from the authors.

*Author contributions.* K. Lakkala analyzed the data and led the manuscript preparation. She was responsible for the QC/QA and processing of Sodankylä data, and supervised the QC/QA and monitoring of Marambio data.

J. Kujanpää implemented the TROPOMI UV data processor and processing system, processed the L2 data and extracted the station overpass files, participated in data analysis and contributed to the writing of the manuscript.

C. Brogniez supervised the monitoring, processing and QC/QA of French spectroradiometer's data and contributed to the writing of the manuscript.

N. Henriot contributed to the sensitivity study regarding the pixel selection and the UVQAV flag.

A. Arola contributed to the development of the TROPOMI UV algorithm, data analysis and the writing of the manuscript.

M. Aun processed Marambio UV data and contributed to the writing of the manuscript.

F. Auriol: Responsible of calibration of the three French spectroradiometers.

A. F. Bais supervised the monitoring, calibration and quality control of THE data.

G. Bernhard: Responsible for QC/QA and processing of Palmer SUV-100 data, and contributed to the writing of the manuscript.

V. De Bock is co-responsible (together with Hugo De Backer) for the QC/QA and processing of Uccle Brewer spectroradiometer data.

M. Catalfamo contributed in calibration of VDA and OHP spectroradiometers.

C. Deroo: Responsible of automatic processing of the three French spectroradiometer's data.

H. Diémoz: Responsible of QC/QA and processing of Aosta data and contributing to the writing of the manuscript.

L. Egli: Responsible for QC/QA and processing of the Davos Brewer spectroradiometer data.

J.-B. Forestier: Technical manager of MAH,ANT,ROD and STD.

I. Fountoulakis: Contribution in QC/QA and processing of Aosta data and contributing to the writing of the manuscript.

K. Garane: Maintenance and calibration of Thessaloniki Brewer.

R. D. Garcia responsible for QC/QA and processing of data of the Izaña BSRN.

J. Gröbner: Responsible for QC/QA of the Davos measurements and contributed to the writing of the manuscript.

S. Hassinen: Overseeing the work in the group and contributed to the TROPOMI data handling.

A. Heikkilä: Responsible for the maintenance of Helsinki Brewer and QC/QA and processing of Helsinki Brewer data.

S. Henderson: Responsible for QC/QA and processing of data of ALI and MEL and contributed to the writing of the manuscript.

G. Hülsen: Responsible for QC/QA and processing of the Davos Qasume spectroradiometer and broadband data and contributed to the writing of the manuscript.

B. Johnsen: Responsible for the QC/QA of the Norwegian UV-network sites and contributed to the writing of the manuscript.

N. Kalakoski contributed to the development of the TROPOMI UV algorithm.

A. Karanikolas contributed in calibration and analysis of THE data.

T. Karppinen contributed to the QC/QA of the Sodankylä Brewer data.

K. Lamy: Responsible for QC/QA and processing of data of MAH, ANT, ROD and STD and contributed to the writing of the manuscript.

S. Leon: Co-Responsible for QC/QA and processing of IZO Brewer spectroradiometer data.

A. V. Lindfors contributed to the development of the TROPOMI UV algorithm and writing of the manuscript.

655 J.-M. Metzger contributed in calibration and maintenance of OPA spectroradiometer.

F. Minvielle contributed in calibration of VDA and OHP spectroradiometers.

H. B. Muskatel: is responsible for the QC/QA and data processing of the measurements from BET, JER and EIL.

T. Portafaix: P.I. of UV-Indien network and responsible for QC/QA of MAH,ANT,ROD and STDR.

A. Redondas: Team leader of Regional Brewer Calibration Center-Europe (RBCC-E). Co-Responsible for QC/QA and processing of IZO
Brewer spectroradiometer data.

R. Sanchez: Responsible in SMN for Marambio UV measurements.

A. Siani: Responsible for QC/QA and processing of ROM data and contributed to the writing of the manuscript.

T. Svendby: Responsible for GUV measurements at Blindern, Andøya and Ny-Ålesund.

J. Tamminen participated in the development of TROPOMI UV data product, discussion of the validation results and manuscript preparation.

*Competing interests.* No competing interests are present.

*Acknowledgements.* The TROPOMI UV-VIS-NIR-SWIR spectrometer on-board EU's Copernicus Sentinel -5P satellite has been developed in cooperation by The Netherlands and European Space Agency. We acknowledge the contribution of the international TROPOMI/Sentinel-5P team, and in particular want to thank KNMI and ESA. We thank the AC SAF project of the EUMETSAT for providing data and/or
670 products used in this paper. We thank the operators of each site for their valuable work in everyday housekeeping of the measurements. We acknowledge Mikko Pitkänen for help with data acquisition. Juha M. Karhu and Markku Ahponen are acknowledged for calibrations of the Brewer measurements in Sodankylä. We thank Patrick Disterhoft and Scott Stierle for providing data of NOAA's UV Monitoring Network. Data from Palmer are from NOAA's UV Monitoring Network data base (https://www.esrl.noaa.gov/gmd/grad/antuv/). UV-Indien program is funded by European Union throught the PO INTERREG V, by the Reunion Island Council and by the French Government. OPAR station
(Observatoire de Physique de l'Atmosphère de La Réunion), and the OSU-R activities are funded by Université de La Réunion and CNRS. Measurements from the French sites are supported by CNES within the TOSCA program, by the Région Hauts-de-France and the Ministère de l'Enseignement Supérieur et de la Recherche (CPER Climibio). GUV measurements at Blindern, Ny-Ålesund and Andøya are funded by the Norwegian Environment Agency. K. Lakkala is supported by the CHAMPS project of the Academy of Finland under the CLIHE program.

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
