# Peer review of "Validation of TROPOMI Surface UV Radiation Product"

_Atmospheric Measurement Techniques, 2020_

## Referee Comment (RC1) · Anonymous Referee #2 · 1 Jul 2020

This is a thorough and well written manuscript detailing comparison of TROPOMI UV products with ground based data from across the globe. There are a few corrections or additions that should be made to provide additional clarity:

Abstract line 13 For compete clarity please state 'relative differences. . .were a little biased towards negative values (i.e. ground-based measurement > satellite data).

Line 40-41 This somehow implies that there is no more chemical ozone depletion. Please rephrase.

Line 53-54 Provide resolution of e.g. OMI UV products to give context to the claim of better resolution.

Line 95-97 Are the changes in ozone product version and aerosol index product version

significant? Please comment in either case.

Section 2.2 The sites are well described, but quantification of the uncertainty in measurements from each site is inconsistent or missing. Ideally all uncertainties would be described in the same way, and added to tables 3 and 4 after the traceability column. Please do what is possible in this respect (at the least provide uncertainties in the text, for those sites currently without).

Please provide a summary statement about the uncertainties in the ground-based data. It is important to know how the ground based data compare as a benchmark for the satellite data comparison. For example, do all instruments that have been compared with the QASUME instrument fall within x% of this world calibration standard? Or, are all expanded uncertainties within y%?

Line 338 Please state which method was then used for the rest of the analysis.

Line 398 Comment briefly on how this improved the relative difference statistics for Davos. For this, and the previous comment, readers should not have to go to supplementary material to find the outcome, only to see the detail.

Results / Conclusion Please provide a summary of comparative results for OMI and TOMS UV products. Specific cases are detailed in the results – mainly those with large relative differences. Please also give the comparators for the 'easier' sites. From what is provided it appears that TROPOMI results are similar to those for OMI – this is important with respect to the final paragraph of the manuscript and the desire for a long time series of satellite derived UV data. Please summarise whether this is the case, or whether significant improvements have been made.

There are multiple minor grammatical errors that can be corrected in editing or permitted by the journal. They do not impact on understanding.

---

## Referee Comment (RC2) · Anonymous Referee #3 · 1 Jul 2020

Review of "Validation of TROPOMI Surface UV Radiation Product" This is a well written review of the TROPOMI UV product with two notable omissions. 1) is a more thorough outline review of the algorithm used, and 2) comparison with the OMI product. Of these, the second is the most important, since a reader mat want to combine the two time series to gain a longterm view of the changes in UV reaching the surface. The description of the algorithm on page 4 may be adequate through references but leaves out a lot of key features. It would be useful to know more about the implementation of cloud transmission through cloud optical depth and how partial cloud coverage within a pixel is handled. The same is true of aerosol absorption calculations. The authors state that the aerosol index is used but give no indication of how the height problem inherent in the aerosol index is resolved or what the algorithm entails. Given the length of this

paper, these are crucial details that could be summarized in an appendix or in the supplement. The comparison with ground-based instruments is very well described. There are problems with comparisons with broad-band instruments Table 4 that are not discussed in this paper. Broad-band instruments do not have a spectral response that matches the erythemal action spectrum used. Figure 5 suggests the difficulty of using broad-band instruments. Instead of scatter plots that clearly indicate problems, time series would be much more revealing of the deficiencies of broad-band analysis, especially the seasonal differences. In contrast, the comparisons with spectrometer type instruments are quite good. While this is not a paper on the quality of ground-based instruments, since the authors included the broad-band results, additional discussions of the problems should be included, or the broad-band comparisons removed. It would be preferable to have additional discussion of broad-band problems. The descriptions of the spectrometer measurements are excellent and form the strongest validation of the TROPOMI estimates. Problems with snow covered conditions are to be expected and are not indicative of problems with TROPOMI. However, the O2 A-band information from TROPOMI can detect clouds over snow and ice and perhaps improve the results. This paper should be published as a valuable reference paper for TROPOMI. Adding comparisons with OMI UV estimates are essential before publication.

---

## Referee Comment (RC3) · Anonymous Referee #1 · 14 Jul 2020

General

This paper is a very useful validation study of the TROPOMI UV product. Because of its unprecedented high spatial resolution of only about 3-5 km, TROPOMI provides a very useful addition to the already existing UV products from satellite instruments like OMI and GOME-2. The quality assessment of UV products at this high resolution is very important for its use.

The validation work performed in this paper is very robust. The used data set is extensive, using 20 months of TROPOMI data and 25 sites at a large range of latitudes. The paper is clearly written and has a straightforward structure. Very good referencing.

Main comments

1. The question naturally arises whether the validation results improve for such a high-resolution satellite instrument like TROPOMI as compared to those of coarser resolution instruments like OMI or GOME-2. At high resolution the specific site conditions are more representatively measured from space: the homogeneity should improve. On the other hand, the cloudiness conditions for larger pixels may be more representative. A comparison of the TROPOMI validation results with OMI and GOME-2 UV validation would be useful. This topic would deserve more attention in discussion and conclusions.

2. Which TROPOMI UV algorithm improvements are needed? Clearly the surface albedo of TROPOMI should be improved and should have a time-component because of the snow variability. Are there more improvements needed as follow from this validation study?

Detailed comments

Abstract, l. 1-5: Those instrument details on TROPOMI do not belong in the abstract.

Abstract: Please mention which UV retrieval algorithm was used.

Abstract l. 13: Please clarify: TROPOMI UV is too low?

l. 72: On the surface albedo data base: which spatial resolution? based on which satellite instrument?

l. 81: Is there a manual (PUM) to explain all the 36 parameters?

l. 372: please clarify on the topic how the UV processor deals with clouds

Caption Figure 5: "Red diamonds…": but such conditions are not all exclusive: sites can be both clear-sky and snow-free. How to indicate that?

l. 425: in function of > as a function of

l. 485: check spelling: homogeneous

---

## Author Comment (AC1) · 8 Sep 2020

Final Author comments

Authors' response to Referee #1, Referee #2 and Referee #3 comments on " Validation of TROPOMI Surface UV Radiation Product" by Kaisa Lakkala et al.

The authors thank the Referees for constructive comments and reply to all comments here below. The answer is structured as follow: (1) comments from Referee, (2) author's response, (3) author's changes in the manuscript.

Referee #1

Main comments

(1) 1. The question naturally arises whether the validation results improve for such a high-resolution satellite instrument like TROPOMI as compared to those of coarser resolution instruments like OMI or GOME-2. At high resolution the specific site conditions are more representatively measured from space: the homogeneity should improve. On the other hand, the cloudiness conditions for larger pixels may be more representative. A comparison of the TROPOMI validation results with OMI and GOME-2 UV validation would be useful. This topic would deserve more attention in discussion and conclusions.

(2) The authors agree with the comment and comparison with OMI and GOME-2 validation results has been added and discussed.

(3) Tables has been added to compare validation results between TROPOMI, OMI and GOME-2. A new section has been added with the following text: "5.1 Comparison with other satellite surface UV products TROPOMI is planned to continue OMI surface UV time series. A detailed comparison analysis of OMI and TROPOMI surface UV product is needed and it is a subject for future study. Many publications have discussed OMI surface UV products, but only few included same sites and same UV parameters. In Tables 7–8 TROPOMI surface UV product validation results are shown together with those from OMI and GOME-2 satellite instrument studies at sites having comparable results. When interpreting the results, one should keep in mind, that each study has different data time period, spatial and time difference, quality criteria and the overpass time of the day vary between satellites. In addition, the pixel size of each satellite instrument is different. These suggest that results depend on actual cloudiness and other atmospheric conditions like aerosols at polluted sites or sites affected by seasonal aerosol or dust events, as well as surface albedo conditions.

[revised manuscript text omitted]

Detailed comments

(1) Abstract, l. 1-5: Those instrument details on TROPOMI do not belong in the abstract.

(2) The authors think that the details are interesting for the readers who want quickly to know what is the paper about: 1) Start date of TROPOMI measurements, 2) Polar orbit and resolution information are important to know when considering the potential application of the data.

(3) No changes in the manuscript.

(1) Abstract: Please mention which UV retrieval algorithm was used.

(2) The TROPOMI UV algorithm (Lindfors et al., 2018).

(3) The following text has been added to the abstract: "The Finnish Meteorological Institute (FMI) is responsible for the development of the TROPOMI UV algorithm and the processing of the TROPOMI Surface Ultraviolet (UV) Radiation Product which includes 36 UV parameters in total."

(1 )Abstract l. 13: Please clarify: TROPOMI UV is too low?

(2) Yes, too low. The text has been clarified.

(3) The text has been changed to: "Generally median relative differences between TROPOMI data and ground-based measurements were a little biased towards negative

values (i.e. satellite data < ground-based measurement), but at high latitudes where non-homogeneous topography and albedo/snow conditions occurred, the negative bias was exceptionally high, from -30% to -65%."

(1) l. 72: On the surface albedo data base: which spatial resolution? based on which satellite instrument?

(2) The surface albedo data is taken from a climatology generated for the OUV algorithm (Kujanpää and Kalakoski, 2015 ) which is provided on a $0.5°\times0.5°$ latitude-longitude grid. It uses the monthly minimum Lambert equivalent reflectivity (MLER) climatology (Herman and Celarier, 1997) for regions and time periods with permanent or negligible snow/ice cover, while elsewhere a climatology from Tanskanen 2004 is used, which better captures the seasonal changes in the surface albedo during the snow/ice melting and formation periods. The following data sets were used to determine the regions and time period with permanent or negligible snow/ice cover: Northern hemispheric monthly snow cover extent data (Armstrong and Brodzik, 2010) from the International Satellite Land-Surface Climatology Project, Initiative II (ISLSCP II) (Hall et al., 2006) together with the monthly masks of maximum sea ice extent derived by the National Snow and Ice Data Center (NSIDC) from the sea ice concentrations obtained from passive microwave data (Cavalieri et al., 1996). The climatology of Tanskanen 2004 is calculated from TOMS 360 nm Lambertian Equivalent Reflectivity (LER) time-series 1979-1992 using the moving time-window method presented in Tanskanen et al. 2003. The data is available in a $1°\times1°$ latitude-longitude grid from http://promote.fmi.fi/MTW_www/MTW.html .

(3) The above text has been added to the manuscript.

New References: Armstrong, R. and Brodzik, M. J.: ISLSCP II Northern Hemi-sphere monthly snow cover extent, ISLSCP Initiative II Collec-tion, in: ISLSCP Initiative II Collection. Data set, edited by:Hall, R. G., Collatz, G., Meeson, B., Los, S., Brown de Col-stoun, E., and Landis, D., available at: http://daac.ornl.gov/ from Oak Ridge

National Laboratory Distributed Active Archive Cen-ter, Oak Ridge, Tennessee, USA, doi:10.3334/ORNLDAAC/982,2010.

Cavalieri, D. J., Parkinson, C. L., Gloersen, P., and Zwally, H.:Sea Ice Concentrations from Nimbus-7 SMMR and DMSPSSM/I-SSMIS Passive Microwave Data, NASA DAAC at theNational Snow and Ice Data Center, Boulder, Colorado, USA,doi:10.5067/8GQ8LZQVL0VL, 1996.

Hall, F. G., Brown de Colstoun, E., Collatz, G. J., Lan-dis, D., Dirmeyer, P., Betts, A., Huffman, G. J., Bounoua, L.,and Meeson, B.: ISLSCP Initiative II global data sets: sur-face boundary conditions and atmospheric forcings for land–atmosphere studies, J. Geophys. Res.-Atmos., 111, D22S01,doi:10.1029/2006JD007366, 2006.

Herman, J. R. and Celarier, E. A.: Earth surface reflectivity clima-tology at 340 nm to 380 nm from TOMS data, J. Geophys. Res.,102, 28003–28011, 1997.

Tanskanen, A., A. Arola, J. Kujanpää, Use of the moving time-window technique to determine surface albedo from the TOMS reflectivity data, In: Proc. SPIE Vol. 4896, p. 239–250, 2003.

Tanskanen, A., Arola, A., and Kujanpää, J.: Lambertian surfacealbedo climatology at 360 nm from TOMS data using movingtime-window technique, in: Proc. XX Quadrennial Ozone Sym-posium, Kos, Greece, 1–8 June 2004, 1159–1160, 2004

(1) l. 81: Is there a manual (PUM) to explain all the 36 parameters?

(2) Yes, The reference of the PUM was added.

(3) The TROPOMI L2 UV product (Kujanpää 2020) contains 36 UV parameters in total (Table 1), including irradiances at four different wavelengths and dose rates for erythemal (Commission Internationale de l'Eclairage, 1998) and vitamin D synthesis (Bouillon et al.,2006) action spectra.

(1) l. 372: please clarify on the topic how the UV processor deals with clouds

(2) For Aosta, which is a challenging mountainous site, the UV product quality parameter flags "rough terrain" which, in the UV algorithm version used for satellite data calculation of this study, sets automatically cloud optical thickness to zero, meaning no clouds. This results in satellite UV data that are too high, especially when heavy clouds are present, which would lead to low UV dose rates measured at the ground. The same applies for Izana which, depending on the TROPOMI pixel position, can be flagged as rough terrain.

(3) Text has been changed to "Large positive biases in TROPOMI UV data occur over these mountainous regions during cloudy conditions when the "rough terrain" quality flag is active and cloud optical depth is set to zero in the UV algorithm."

(1) Caption Figure 5: "Red diamonds...": but such conditions are not all exclusive: site scan be both clear-sky and snow-free. How to indicate that?

(2) Here clear-sky means that the cloud optical depth retrieved from the first LUT of the TROPOMI UV algorithm is less than 0.5. It doesn't tell about the actual conditions at the measurement site. To distinguish between snow and snow-free, the input of TROPOMI albedo is used, which is based on albedo climatology, and neither tells about actual albedo conditions of the site. This means that if TROPOMI data with COD<0.5 (in the first manuscript version called "clear sky") agree well with ground-based measurements, most probably the albedo climatology is representative for actual surface albedo conditions, assuming there is no problem with other input parameters (e.g. aerosols). The authors are aware that the wording "clear sky" can be misleading and it has been changed in the whole manuscript to "cloudfree".

(3) The following text has been added to the manuscript:" Results for the cloudfree datasets, with cloud optical depth input parameter of the TROPOMI UV algorithm lower than 0.5, were also included in the plot. Cloudfree criteria is not reflecting actual cloudiness conditions, but the cloud optical depth retrieved from the first LUT in the TROPOMI UV algorithm. If a perfect agreement between satellite and ground-based

data is found, most probably also the surface albedo climatology represents actual surface albedo conditions and the aerosol climatology actual aerosol conditions. "

(1) l. 425: in function of > as a function of

(2) Text changed as suggested.

(3) Text changed as suggested.

(1) l. 485: check spelling: homogeneous

(2) Text changed as suggested.

(3) Text changed as suggested.

Referee #2

(1) Abstract line 13 For compete clarity please state 'relative differences...were a little biased towards negative values (i.e. ground-based measurement > satellite data).

(2) Text changed as suggested.

(3) Text changed as suggested.

(1) Line 40-41 This somehow implies that there is no more chemical ozone depletion. Please rephrase.

(2) The text has been rephrased.

(3) The text has been changed to: "The international Montreal Protocol was signed in 1987 to protect the ozone layer by phasing out the production of ozone-depleting substances (ODS). As a result, the ozone layer is now starting to recover (WMO, 2018). However, the removal process of ODS will take several decades and UV levels at the ground will therefore remain elevated for the foreseeable future (Petkov et al., 2014; Fountoulakis et al., 2020)."

(1) Line 53-54 Provide resolution of e.g. OMI UV products to give context to the claim

of better resolution.

(2) The OMI pixel size is 13 km× 24 km at nadir.

(3) The text has been changed to: "The ground resolution of the UV product was 7.2x3.5 km$^2$ at nadir until 6 August 2019, and is now 5.6x3.5 km$^2$, while the OMI pixel size was 13x24 km$^2$ at nadir.".

(1) Line 95-97 Are the changes in ozone product version and aerosol index product version significant? Please comment in either case.

(2) Changes in version numbers do not significantly impact the surface UV product. However, there are signs of degradation in the UV solar irradiance measurement of TROPOMI. We do not see any trend in our cloud optical depth retrievals using the 354 nm reflectance, but further analysis is needed in any UVtrend study."

(3) The following text has been added to the manuscript: "Changes in version numbers do not significantly impact the surface UV product. However, there are signs of degradation in the UV solar irradiance measurement of TROPOMI (Rozemeijer and Kleipool, 2019). We do not see any trend in our cloud optical depth retrievals using the 354 nm reflectance, but further analysis is needed in any UV trend study."

Reference: Rozemeijer, N. C. and Kleipool, Q., S5P Mission Performance Centre Level 1b Readme, S5P-MPC-KNMI-PRF-L1B, issue 2.2.0, 31.10.2019, 2019, available at http://www.tropomi.eu/sites/default/files/files/publicSentinel-5P-Level-1b-Product-Readme-File.pdf

(1) Section 2.2 The sites are well described, but quantification of the uncertainty in measurements from each site is inconsistent or missing. Ideally all uncertainties would be described in the same way, and added to tables 3 and 4 after the traceability column. Please do what is possible in this respect (at the least provide uncertainties in the text, for those sites currently without).

(2) The authors agree with the Referee and comparison against QASUME and expanded uncertainties are included in Tables 3 and 4 when available.

(3) The following text has been changed: "Many of the spectroradiometers have participated in on- site quality assurance of spectral solar UV measurements performed by the traveling reference spectroradiometer QASUME since 2002 (Gröbner et al., 2005), and the average offset of all instruments is within ±5% from the reference instrument with a diurnal variability typically less than 5%. The reports of the site visits can be found at https://www.pmodwrc.ch/en/world- radiation-center-2/wcc-uv/qasume-site-audits/ and the comparison results of the latest QASUME comparisons are shown in Table 3. In addition, available estimates of expanded uncertainties of ground-based measurements are shown in Tables 3 and 4. The expanded uncertainties of spectroradiometers and broadband / multiband radiometers are less than 6% and less than or equal to 9%, respectively."

(1) Please provide a summary statement about the uncertainties in the ground-based data. It is important to know how the ground based data compare as a benchmark for the satellite data comparison. For example, do all instruments that have been compared with the QASUME instrument fall within x% of this world calibration standard? Or, are all expanded uncertainties within y%?

(2) The authors agree with the Referee and a summary statement about the uncertainties is added.

(3) Please see answer to the question above.

(1) Line 338 Please state which method was then used for the rest of the analysis.

(2) The text has been modified to include the method used in the rest of the analysis: 1) Each TROPOMI pixel was treated as individual measurement.

(3) The modified text is now: "The spatial resolution of TROPOMI data is very high compared to older generation satellite instruments. This leads to huge amount of data and at most sites several satellite pixels fulfilling the selection criteria were colocated

with the same ground-based measurement. For example, at high latitudes, this increased the number of data to include more than 5 pixels for each overpass. Thus, the sensitivity of the results was studied by comparing three different data selection methods for Villeneuve d'Ascq measurements: 1) Each TROPOMI pixel was treated as individual measurement, 2) the pixel nearest of the site was chosen, 3) the average of the TROPOMI pixels meeting the chosen limitations (time difference, SZA, altitude, distance) was used. The results did not differ significantly between the methods, and in this study the results were calculated for each pixel separately. Results are shown in the Fig. S13 and Table S6 of the Supplement material."

(1) Line 398 Comment briefly on how this improved the relative difference statistics for Davos. For this, and the previous comment, readers should not have to go to supplementary material to find the outcome, only to see the detail.

(2) Comment on the changes in statistics has been added. The results of the "pixel test" are addressed in the response of the previous comment.

(3) The corresponding chapter if now: "The effect of taking into account quality flags was evaluated for the site of Davos. Data for which the quality value number UVQAV was less than 0.5 were excluded (see Section 2.1 for explanation of UVQAV). This removed e.g., unreliable values when the cloud optical depth was set to 0 due to the flagging. Indeed, as mentioned in Section 2.2, Davos is a mountainous site with heterogeneous albedo during the winter. Setting a limit of 0.5 for the UVQAV, results in removing satellite observations with at least two of the following warnings: "rough_terrain", "alb_hetero" or "clearsky_assumed". This procedure reduced the number of data points by around half, and removed most data points where satellite estimates exceed the ground measurements. This resulted in a shift of median relative differences towards more negative values: From -24% to -57% and from -6% to -13 % for snow cover and snow free conditions, respectively. The statistics and scatter plots of the study are shown in Supplement material Table S7 and Fig. S14."

Results / Conclusion

(1) Please provide a summary of comparative results for OMI and TOMS UV products. Specific cases are detailed in the results – mainly those with large relative differences. Please also give the comparators for the 'easier' sites. From what is provided it appears that TROPOMI results are similar to those for OMI – this is important with respect to the final paragraph of the manuscript and the desire for a longtime series of satellite derived UV data. Please summarise whether this is the case, or whether significant improvements have been made.

(2) The authors agree with the comment and comparison with OMI and GOME-2 validation results has been added and discussed, and a summary sentence with TOMS results have been added to the Discussion section.

(3) Please see changes in the manuscript from the first answer to Referee #1.

Referee #3

(1) A more thorough outline review of the algorithm used

(2) The used TROPOMI UV algorithm has already been published in Lindfors et al. 2018, so the authors think that a restricted amount of details is sufficient for this paper. Details which are important for discussion on results have been included. As suggested more details on cloud optical depth retrieval and aerosol correction are included. Also the albedo climatology has been discussed in more details.

(3) Albedo related changes: See response to Referee#2.

Cloud optical depth: The paragraph describing the cloud optical depth retrieval has been modified to:"The TROPOMI UV algorithm is based on two pre-computed lookup tables (LUT) in order to save computing time compared to explicit radiative transfer calculations. The first LUT is used to retrieve the cloud optical depth from the measured 354 nm reflectance using SZA, viewing zenith angle, relative azimuth angle, surface pressure and surface albedo as other inputs. Details on the cloud optical depth retrieval

can be found in Sect. 3.3 of Lindfors et al, 2018s. The measured 354 nm reflectance together with the angles and surface pressure are obtained from the TROPOMI L2 aerosol index (AI) product (Stein et al, 2018) as they are used for the calculation of the AI product. The LUT was pre-generated by radiative transfer calculations. The reflectance and 354 nm was calculated using different combinations of cloud optical depth, SZA, viewing zenith angle, relative azimuth angle, surface pressure and surface albedo. The outcome is a LUT from which the cloud optical depth can be retrieved when all other input parameters are known. For radiative transfer calculations, a homogeneous water cloud layer is considered at 1-2 km height in the atmosphere. Thus, the retrieved cloud optical depth can be considered to be an effective optical depth for the whole satellite pixel which best corresponds to the measured 354 nm reflectance. 3D effects due to partial cloudiness are ignored. "

Aerosol absorption: The following has been added to the text of the manuscript:"The correction for absorbing aerosols follows the approach developed earlier to the OMI algorithm (Arola et al. 2009). It is based on aerosol absorption optical depth (AAOD), which is taken from the monthly aerosol climatology by Kinne et al. (2013). The correction factor and its dependence on AAOD was first suggested by Krotkov et al. 2005 and applied in Arola et al. 2009."

(1) Adding comparison with the OMI product.

(2) The authors think that a detailed comparison between TROPOMI and OMI is a subject for another study. However discussion on comparison of published validation results of different satellite instruments has been added as a new section to the manuscript.

(3) Please see the first answer to Referee #1.

(1) There are problems with comparisons with broad-band instruments Table 4 that are not discussed in this paper. Broad-band instruments do not have a spectral response that matches the erythemal action spectrum used. Figure 5 suggests the difficulty of

using broad-band instruments. Instead of scatter plots that clearly indicate problems, time series would be much more revealing of the deficiencies of broad-band analysis, especially the seasonal differences.

(2) The broadband instruments have a spectral response close to the erythemal action spectrum, but it is true that it is not perfectly the same. This difference is taken into account during the calibration procedure of broad band instruments, as the calibration coefficient is provided as a function of SZA and total ozone. Data from broadband instruments can agree within $\pm 2\%$ with measurements of well- calibrated spectrora- diometers (Hülsen et al. 2008 and Hülsen et al. 2020). The authors don't see that Figure 5 suggest difficulties using broadband instruments. The broadband instruments used in the study are from Davos (DBB), the Israeli sites (BET, JER, EIL), Australian site (ALI) and the Indian Ocean sites (MAH, ANT, SDT, ROD). From those, the Is- raeli sites and the Australian site don't show seasonal dependence. The discrepancies between TROPOMI and ground-based measurements in Davos are similar to those found by performing the 310 nm irradiance comparison against the spectral instrument (DAV), where the median relative differences are -39% (snow cover) and -8% (snow free) and -38% (snow cover) and -5% (snow free) for DBB and DAV, respectively. The clear seasonal dependency is due to problems of TROPOMI surface UV product for non-homogenous topography sites, not because of problems with broadband instru- ments. For the Indian Ocean sites, there is no clear reason for the underestimation of TROPOMI UV at extreme UV levels. The biggest differences is seen for three Indian Ocean sites and the possible explanation is discussed in the manuscript: " The same applies for the impact of cloudiness when clouds are non-uniformly located around the site due to topography or changes in surface (e.g. sea/ground). For example the site itself is free from clouds but there can be small cumulus clouds at the edge of the pixel which increase the reflection towards the satellite. In that case the TROPOMI would most probably underestimate the UV irradiance as the small fraction of clouds is considered as cloudiness of the whole pixel. Under scattered clouds, the UV radi- ation at the surface can be larger than during clear skies (Calbó et al., 2005; Jégou

et al., 2011). This phenomenon occurs when the direct radiation from the Sun is not obstructed and additional radiation is scattered by clouds to the radiometer at the surface. The TROPOMI algorithm does not consider these situations, resulting UV levels that are too low. This phenomenon is likely one reasons for the underestimation found in the TROPOMI UV dose rates during high UV levels at tropical sites." Plots of time series will not help to identify the reason of the discrepancies, as the self-evident noise due to how TROPOMI algorithm handle cloudiness (scattered cloudiness not detected by TROPOMI) will mask other reasons.

(3) Text has been added about the uncertainties related to broadband measurements and uncertainties have been added to the manuscript (Table 4). Please see answer to Referee #2.

New Reference: Hülsen, G., Gröbner, J., Bais, A., Blumthaler, M., Diémoz, H., Bolsée, D., Diaz, A., Fountoulakis, I., Naranen, E., Schreder, J., Stefania, F., and Guerrero, J. M. V.: Second solar ultraviolet radiometer comparison campaign UVC-II, Metrologia, 57, 035 001, https://doi.org/10.1088/1681-7575/ab74e5, https://doi.org/10.1088%2F1681-7575%2Fab74e5, 2020.

(1) Problems with snow covered conditions are to be expected and are not indicative of problems with TROPOMI. However, the O2 A-band information from TROPOMI can detect clouds over snow and ice and perhaps improve the results.

(2) The authors thank for the suggestion.

[revised manuscript text omitted]